



# Validation of demographic equilibrium theory against tree-size distributions and biomass density in Amazonia

Jonathan R Moore[1], Arthur P K Argles[1], Kai Zhu[2], Chris Huntingford[3], and Peter M Cox[1]

[1]College of Engineering, Mathematics and Physical Sciences, University of Exeter, Exeter, Devon EX4 4QF, UK
[2]Department of Environmental Studies, University of California, Santa Cruz, California 95064, USA
[3]Centre for Ecology and Hydrology, Wallingford, OXON OX10 8BB, UK

**Correspondence:** Jonathan Moore (j.moore3@exeter.ac.uk)

**Abstract.** Understanding the relative abundance of trees of different sizes is an important part of predicting the response of forests to changes in climate, land-use and disturbance events. Two competing theories of forest size-distributions are demographic equilibrium theory (DET), based on scaling of mortality and growth with size, and metabolic scaling theory (MST), based scaling size with metabolic rates and how trees fill space. Recently, it was shown that for US forests DET is

a much better model than MST, even using the same growth scaling with size. Studies comparing DET and MST have so far focused on trunk diameter, but tree mass and the associated forest mass per unit area (biomass density) are much more relevant to climate. In this study, we extend by fitting both DET and MST to mass data for the Amazon rainforest. The conversion via allometry from trunk diameter data to mass leads to an artefact in the mass distribution, which can be corrected by excluding smaller trees. We derive equations to calculate the total forest biomass density from the mass distribution equation,

for both models, and these can be used as an indicator of goodness of model fit to the data. The models were fitted to the data, using Maximum Likelihood Estimation, at the forest plot, regional and continental scale. The fits for both diameter and mass demonstrate that MST is rarely a good fit for Amazon size-distributions and that DET is much better and can estimate biomass density, at the forest plot scale, with a mean error of 6% (10% if DET allometry fixed to MST) of its true value, compared to 139% for MST. The median of the fitted growth scaling power for all the 124 plots is very close to the MST allometry values,

implying MST allometry is a mean scaling, around which smaller forest plots cluster. At the larger regional scale, the error in the biomass density estimate of DET reduces to 2% or less and it is less than 1% for the whole continent. This suggests that models based on DET, such as the relatively simple Robust Ecosystem Demography model (RED), are a good basis for a next-generation dynamic global vegetation model, and that Amazonian forests remain close to demographic equilibrium on large-scales, despite climate change and significant anthropogenic disturbance.



# 1 Introduction

The modelling of the abundances of various tree sizes in tropical forests such as the Amazon is important in efforts to improve understanding of land-climate feedbacks and hence anthropogenic climate change. This is because the fluxes of $CO_2$ between land and atmosphere are sensitive to climate (Feldpausch et al., 2016; Gatti et al., 2014) and so lead to possibly significant
5 climate-land feedbacks, such as those predicted for the Amazon (Cox et al., 2000; Brienen et al., 2015).

 Earth System models (ESMs) are used to model climate, but currently have a large range of uncertainty in the prediction of the land carbon sink, with as much as 500 GtC uncertainty by 2100 for 1% increase in $CO_2$ emissions per year (Friedlingstein et al., 2014). This has an impact on the predictions of how much emissions need to be reduced to keep global warming within a certain level.

10 These issues have led to the development of more advanced Dynamic Global Vegetation Models (DGVMs), used within ESMs, to better represent vegetation processes (Sitch et al., 2015; Fisher et al., 2018). One of the key advances has been the inclusion of tree size-distributions, which allows better representation of land-use change and recovery from disturbance. These more recent DGVMs broadly consist of two different approaches based on representing size either via individual based models (Shugart et al., 2018) or using cohort-based ecosystem demography models (Moorcroft et al., 2001; Longo et al., 2019).

15 DGVMs though also need to balance additional complexity against practical considerations of usability, and computer execution time and memory usage. Key issues in the usability of complex numerical models are the understanding of the effect of many model parameters and dependence on initial conditions (Moore et al., 2018). One, often overlooked, solution is simple analytical models that only represent key features of a numerical model, but allow a complete view of the effect parameters have on their behaviour. These analytical models can, therefore, be used to aid understanding of more complex models and
20 also can have practical uses when validating models against data or using equilibrium solutions to aid model initialisation. This approach has been used with the new DGVM called the Robust Ecosystem Demography model (RED, Argles et al., in preparation). The analytical equilibrium solution, as well as being used for initialisation, can be extended to allow RED to estimate implicit disturbance rates based on average vegetation coverage and growth.

 Moore et al. (2018) compared simple trunk diameter size-distribution models to US forest inventory data. This approach
25 showed that even in areas like the US that are not yet completely at equilibrium, the size-distribution could be modelled well on the larger scales that are important for DGVMs. We now extend this approach by firstly comparing various analytical models to forest inventory data for the Amazon, and secondly using mass as well as trunk diameter data. These are two key extensions, as the Amazon is one of the largest pools of land carbon on the planet (Feldpausch et al., 2012) and using mass data allows this approach to test the ability of this modelling approach to accurately predict biomass density.

30 As in our previous study (Moore et al., 2018), we use the Demographic Equilibrium Theory (DET) model (Muller-Landau et al., 2006b) and the Metabolic Scaling theory (MST) model (West, 1997; West et al., 2009). MST has two assumptions of interest; that of how trees fill space and of the allometric scaling of tree growth rate with tree size. MST assumes that trees of varying sizes fill space in such a way that the size-distribution scales with trunk diameter $D$ as $D^{-2}$ and that they have power-law growth rate scaling as $D^{1/3}$. Our previous study tested only the space-filling assumption as the DET model we used





also assumed the same growth scaling as MST. In this study we test two versions of DET, one with the MST growth scaling and another where the assumed power law scaling of growth has an exponent that is also allowed to vary as a fitting parameter.

So this study will test which of the models is the best choice for trunk diameter distributions, for mass distributions and for also for total biomass density prediction for the Amazon.

## 2 Theory

### 2.1 Demographic Equilibrium Theory (DET)

The distribution of tree sizes in a forest can be understood in terms of how the growth and mortality of the trees vary with tree size (Kohyama et al., 2003; Coomes et al., 2003; Muller-Landau et al., 2006b). For a given size class (i.e. range of tree size), then trees smaller than that range will grow into it increasing the abundance. Conversely the abundance will decrease as trees in that range grow out or die. The balance of growth and mortality will determine whether the abundance of a size class is increasing, decreasing or if it is in demographic equilibrium (Van Sickle, 1977). On a whole forest scale there is a further balance between the rate of seedling recruitment from seeds (lower boundary condition) and the whole forest mortality. Again this balance will determine if the forest as a whole is gaining or losing both mass and/or abundance.

The governing equation for this process is variously known as the one dimensional drift or continuity equation (Van Sickle, 1977), the Kolmogorov forward or the Fokker-Planck equation with the second order term omitted (Kohyama, 1991) and the Van Foerster equation (Von Foerster, 1959): -

$$\frac{\partial n(D,t)}{\partial t} + \frac{\partial}{\partial D}\left(n(D,t)\,g(D,t)\right) = -\gamma(D,t)\,n(D,t) \tag{1}$$

where $n$ is the size-distribution (tree density per size class) in Trees cm$^{-1}$ ha$^{-1}$ in terms of tree trunk diameter $D$ in cm, trunk diameter growth rate $g$ in cm year$^{-1}$, $\gamma$ is the mortality rate per year and time $t$ in years.

It was shown (Kohyama et al., 2003) that for an unchanging, equilibrium size-distribution, this equation can be integrated as follows: -

$$\int\limits_{n_L}^{n} \frac{dn}{n} = \int\limits_{D_L}^{D} \frac{1}{g(D)}\left[\frac{dg(D)}{dD} + \gamma(D)\right] dD \tag{2}$$

where $n_L$ is the value of $n$ at the lower boundary $D_L$, which for forest inventory data is the minimum sampling size (in this study 10 cm).

This equation can be solved to give an exact solution for simplifying assumptions of size-independent mortality $\gamma(D) = \gamma$ and power law growth rate $g(D)$ in cm per year

$$g(D) = g_1 D^{\phi} \tag{3}$$





where $g_1$ is a constant with the same value as the growth rate for a tree with trunk diameter of 1 cm. The solution (Muller-Landau et al., 2006b; Lima et al., 2016; Moore et al., 2018) for the size distribution is then the Left-Truncated Weibull Distribution (LTWD)

$$n(D) = n_L \left( \frac{D}{D_L} \right)^{-\phi} \exp \left[ \frac{\mu_1}{1-\phi} \left( D_L^{1-\phi} - D^{1-\phi} \right) \right], \phi \neq 1 \tag{4}$$

where $\mu_1 = \gamma/g_1$ is the mortality to growth ratio at $D$=1 cm.

This solution is also applicable for other size variables such as tree dry mass $m$ in kg: -

$$n(m) = n_L \left( \frac{m}{m_L} \right)^{-\phi_m} \exp \left[ \frac{\mu_{m,1}}{1-\phi} \left( m_L^{1-\phi_m} - m^{1-\phi_m} \right) \right], \phi_m \neq 1 \tag{5}$$

where $m_L$, $\mu_{m,1}$ and $\phi_m$ are the mass equivalents of $D_L$, $\mu_1$ and $\phi$.

The LTWD distribution has been shown to be a good description of tree trunk diameter distributions in a variety of tropical forests (Muller-Landau et al., 2006b; Lima et al., 2016) and in temperate forests in the US over larger scales (Moore et al., 2018). When these distributions are fitted to data then they can have both parameters $\phi$ and $\mu_1$ as fitting parameters or just fit $\mu_1$ and fix $\phi$ to the values used in MST allometry (Niklas and Spatz, 2004; West et al., 2009) of $\phi = 1/3$ and $\phi_m = 3/4$.

## 2.2 Total Biomass Density for DET

The total biomass density (kg of dry tree mass per hectare) of the LTWD tree mass distribution can be obtained by integrating Eq. (5) in terms of mass, between the lower boundary $m_L$ and infinity: -

$$M_{L \to \infty} = \int_{m_L}^{\infty} m \, n(m) dm = n_L m_L^{\phi_m} \frac{\exp(x \mu_{m,1} m_L^{1/x})}{\mu_{m,1}(x \mu_{m,1})^x} \Gamma(x+1, x \mu_{m,1} m_L^{1/x}) \tag{6}$$

where $\Gamma$ is the upper incomplete Gamma Function, $x = 1/(1-\phi_m)$.

As real forests do not satisfy the assumption of infinite maximum size tree, this can lead to errors in the calculated biomass density. A correction to this can be found, in terms of $m_{\max}$ the largest tree mass in the distribution: -

$$M_{L \to max} = \int_{m_L}^{m_{\max}} m \, n(m) dm = \int_{m_L}^{\infty} m \, n(m) dm - \int_{m_{\max}}^{\infty} m \, n(m) dm \tag{7}$$

In cases where $m_{\max}$ is both large and much larger than $m_L$ then there will be little difference between Eq. (6) and Eq. (7). $m_{\max}$ is a somewhat arbitrary function of the sample size, due to large trees being statistically rare, meaning the infinite upper bound solution Eq. (6) is expected to perform better for larger sample sizes.





### 2.3 Metabolic Scaling Theory (MST)

Metabolic scaling theory is a theory of scaling of organisms with size, based on theories of metabolism, physics and chemistry (West, 1997; Muller-Landau et al., 2006a). This theory uses the predictions of the scaling of individuals to predict the larger scale patterns and structure of populations and communities. For forests this is in the form of using the scaling of photosynthesis of trees and the vascular structures that transport water to predict individual scaling. This is then combined with assumptions about how trees fill space to describe the expected forest size-distribution (West et al., 2009). This leads to a power law distribution for trunk diameter: -

$$n(D) = n_L \left( \frac{D}{D_L} \right)^{-2} \tag{8}$$

and for mass the distribution is almost identical

$$n(m) = n_L \left( \frac{m}{m_L} \right)^{-11/8} \tag{9}$$

### 2.4 Total Biomass Density for MST

The MST equations also enable the calculation of biomass density (kg of dry tree mass per hectare). In this case only the finite upper bound of $m_{\max}$ can be used as the solution goes to infinity as the upper bound goes to infinity.

$$M_{L \to max} = \int_{m_L}^{m_{\max}} m\,n(m)\,dm = \frac{8 n_L m_L^{11/8}}{5} \left[ m_{\max}^{5/8} - m_L^{5/8} \right] \tag{10}$$

### 3 Methods

#### 3.1 Forest inventory data

The tree census data used in this study is from the public access permanent sample plots of the RAINFOR (Peacock et al., 2007) network. RAINFOR provides a systematic framework for long-term monitoring of the Amazon. The RAINFOR data is stored on the ForestPlots database (https://www.forestplots.net). This database stores measurements (stem diameter, species ID, recruitment, growth, and mortality) of individual trees from hundreds of locations, taken using standardised techniques to allow the behaviour of tropical forests to be measured, monitored and better understood (Lopez-Gonzalez et al., 2011).

We selected 124 open access forest plots (Fig. 1) classified as mixed forest and old-growth to most closely match the model assumptions of forests undisturbed by human interference and approximating to equilibrium demography. All measurements below a trunk diameter of 10 cm trunk diameter were excluded giving consistent left-truncation point to the distributions of 10 cm. Plots with a large proportion of measurements below 10 cm were not included in the 124 selected plots.



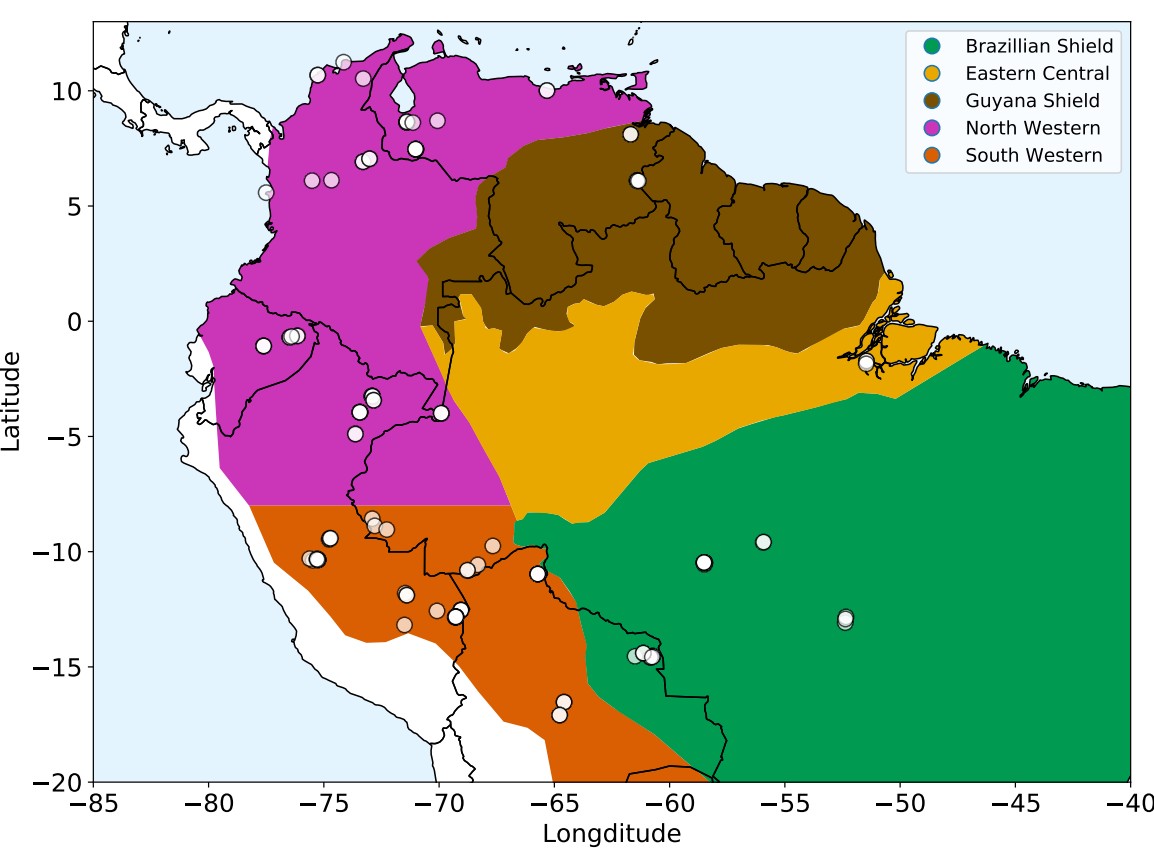

**Figure 1.** Amazonian Allometric Regions. Each region, shown by the coloured areas, is defined by geography, rainfall and soil substrate. White circles show location of the forest plots used. The two western regions share common allometry but are split based on rainfall seasonality for analysis purposes.





**Table 1.** Coefficients for Eq. (11) from Feldpausch et al. (2012)

| Region | $a_h$ | $b_h$ | $c_h$ |
|---|---|---|---|
| All S.America | 42.574 | 0.0482 | 0.8307 |
| Western Regions | 46.263 | 0.0876 | 0.6072 |
| Brazilian Shield | 227.35 | 0.0139 | 0.555 |
| Guyana Shield | 42.845 | 0.0433 | 0.9372 |
| Eastern-Central | 48.131 | 0.0375 | 0.8228 |

## 3.2 Calculating Dry Tree Mass from Trunk Diameter

The open access plots of the Amazon RAINFOR dataset consists only of trunk diameter values. To estimate the tree mass, the methodology developed by Feldpausch et al. (2012) was used. In that study two functional forms (with and without height) were tested against destructively sampled mass data (trees carefully measured then cut down and weighed) to find ones which best estimated mass from trunk diameter. It was found that mass estimation accuracy doubled when including height, even if the height had in turn been estimated from trunk diameter. Out of three choices of height functional form (power law, Weibull-H and exponential), Feldpausch et al. (2012) found to the Weibull-H form Eq. (11) to be the best at estimating mass across multiple size classes. The height $H$ in metres is then: -

$$H = a_h(1 - \exp(-b_h D^{c_h})) \tag{11}$$

with the coefficients varying geographically between defined allometric regions (see Table 1 and Figure 1).

The regions were defined by geography and substrate origin (Feldpausch et al., 2012). Western Amazonia (Columbia, Ecuador and Peru) being recently weathered Andean deposits, the geologically old Brazilian Shield to the south (Bolivia and Brazil), Guyana Shield on the northern side of the Amazonia Basin (Guyana, French Guyana and Venezuela) and Eastern Central Amazonia (Brazil) consisting of sedimentary substrates originating from the other regions. The western region from Feldpausch et al. (2012) was split along latitude of -8 degrees based on rainfall seasonality (Fauset et al., 2015). These two western still retain a common height allometry but are split for analysis.

The mass function (kg), when height was included as one of the parameters used was: -

$$M = e^a(\rho_w D^2 H)^b \tag{12}$$

where the parameters are universal across all regions with values $a = -2.9205$ and $b = 0.9894$. The function was from Feldpausch et al. (2011) and the parameters were estimated in Feldpausch et al. (2012).

The wood specific gravity $\rho_w$ was obtained from the Dryad Global Wood Density Database https://doi.org/10.5061/dryad.234/1 (Chave et al., 2009; Zanne et al., 2009). For each tree measurement the $\rho_w$ value used was for that species from the closest





available region. Where the species data was unavailable or the species of the measurement had not been recorded then the $\rho_w$ value of the Genus was used, based on an average of all trees in the Dryad database in that Genus. Trees without Genus data were estimated from Family data, and any remaining measurements where the $\rho_w$ was still unknown were set to the average $\rho_w$ of the trees in that same forest plot with known $\rho_w$ values.

## 3.3 Fitting methodology

As in our previous study (Moore et al., 2018), Maximum Likelihood Estimation (MLE) was used to find the parameters that give the best fit for both the Left-Truncated Weibull, derived from DET (DET-LTWD) and Metabolic Scaling Theory (MST) distributions. MLE is an effective method for parameter fitting of forest size distributions (Taubert et al., 2013; White et al., 2008).

Maximising the log-likelihood $L$ results in a more numerically tractable summation of terms rather than a product. $L$ in terms of $f(D)$ the probability distribution function (pdf) is then

$$L = \sum_i \ln(f(D_i)) \tag{13}$$

where $D_i$ is tree trunk diameter data point $i$ in the dataset.

The data was fitted both by plot, by allometric region (an aggregated dataset of all plots in that region), by country (again

aggregation of plots) and for all the data, from all 124 plots, grouped together as one large dataset. This allows both the study of the individual plots and the larger scale patterns across South America.

### 3.4 Maximum Likelihood Estimation (MLE) for Demographic Equilibrium Theory (DET)

The probability density function (pdf) $f(D)$ for the DET-LTWD, in terms of tree trunk diameter $D$ and minimum tree size $D_L$ is related to the number density distribution $n(D)$ (Eq. 4): -

$$f(D) = \frac{A}{N} n(D) = \mu_1 D^{-\phi} \exp\left[\frac{\mu_1}{1-\phi}\left(D_L^{1-\phi} - D^{1-\phi}\right)\right], \phi \neq 1 \tag{14}$$

where $N$ is the total number of trees in the dataset being fitted, $\phi$ is the growth scaling power from Eq. (3) and $A$ the area of the plots containing the trees sampled in the dataset. This equation is equivalent to the standard form of the LTWD

$$f(D) = \frac{c}{\lambda}\left(\frac{S}{\lambda}\right)^{c-1} \exp\left[\left(\frac{D_L}{\lambda}\right)^c - \left\{\frac{D}{\lambda}\right\}^c\right] \tag{15}$$

where $c = 1 - \phi$ and $\lambda = \left[\frac{c}{\mu_1}\right]^{1/c}$.

We fit DET-LTWD twice, once with both parameters $\phi$ and $\mu_1$ allowed to vary as fitting parameters and secondly with the growth scaling parameter $\phi$ fixed to the MST allometry values ($\phi = 1/3$ and $\phi_m = 3/4$, see Niklas and Spatz (2004) and West





et al. (2009)). Fixing $\phi$ means we have a DET-LTWD model following just one assumption of MST (the allometry) and so acts as way of comparing the effect of the second MST assumption of space-filling when comparing DET-LTWD and MST fits.

### 3.4.1 One Parameter Fit

For this situation, where we are only aiming to find the parameter $\mu_1$ and $\phi$ is assumed, then MLE can be solved analytically
(Kizilersu et al., 2016)

$$\mu_1 = \frac{c}{\left(\overline{D^c} - D_L^c\right)} \tag{16}$$

where $c = 1 - \phi$. The equations are the same for tree mass, just with the symbols appropriately substituted ($m$ for $D$ etc).

### 3.4.2 Two Parameter Fit

For the two parameter case, where both $\phi$ and $\mu_1$ are fitted, then we calculate the Log-Likelihood $L$ as follows

$$L = N\left(\ln\mu_1 + \mu_1\frac{D_L^c}{c}\right) - \frac{\mu_1}{c}\sum_i D_i^c + (c-1)\sum_i \ln D_i \tag{17}$$

Substituting Eq. (16) into Eq. (17) creates a function only of $c$ and therefore $\phi$. This allows minimisation of $-L$ in terms of $\phi$ by using Brent's bounded algorithm (Brent, 1973). Once the parameters $\phi$ and $\mu_1$ that gives the maximum $L$. Once the parameters $\mu_1$ and $\phi$ are estimated, then this allows $n_L$, the tree density per size class at $D_L$, to be obtained from these parameters and the total number of trees $N$ and the plot area $A$, which are known from the data. This can derived by integrating
the equation for $n$ (Eq. 4), to give: -

$$\frac{N}{A} = \int\limits_{D_L}^{D_{\max}} n(D)dD = \frac{n_L\, D_L^\phi}{\mu_1}\left[1 - \exp\left[\frac{\mu_1}{c}(D_L^c - D_{\max}^c)\right]\right] \tag{18}$$

and noting that the observed number of trees is identical to the integral, we get:-

$$n_L = \left(\frac{N}{A}\right)\frac{\mu_1}{D_L^\phi}\frac{1}{1 - \exp\left[\frac{\mu_1}{c}(D_L^c - D_{\max}^c)\right]} \tag{19}$$

where $c = 1 - \phi$ and $D_{\max}$ is the largest tree size in the dataset. For this study it was found that as $D_{\max} >> D_L$ for most cases
$n_L$ could assumed to be: -

$$n_L \approx \left(\frac{N}{A}\right)\frac{\mu_1}{D_L^\phi} \tag{20}$$

Again, the equations are the same for tree mass, just with the symbols appropriately substituted ($m$ for $D$ etc).





### 3.5 Maximum Likelihood Estimation (MLE) for Metabolic Scaling Theory

From the equation for number density $n$ (Eq. 8) the pdf for MST is

$$f(D) = n(D)\left(\frac{A}{N}\right) = \frac{D_L}{\left[1 - \left(\frac{D_L}{D_{\max}}\right)\right]} D^{-2} \tag{21}$$

where $D_{\max}$ is the largest tree size in the dataset. As all the quantities are known then there are no free parameters to fit and all

that needs to be done is calculate $n_L$, the tree density per size class at $D_L$: -

$$n_L = \frac{(N/A)}{D_L \left[1 - \left(\frac{D_L}{D_{\max}}\right)\right]} \tag{22}$$

Similarly the MST pdf for mass from Eq. (9) is: -

$$f(m) = n(m)\left(\frac{A}{N}\right) = \frac{3m_L^{3/8}}{8\left[1 - \left(\frac{m_L}{m_{\max}}\right)^{3/8}\right]} m^{-11/8} \tag{23}$$

and for $n_L$: -

$$n_L = \frac{3(N/A)}{8m_L \left[1 - \left(\frac{m_L}{m_{\max}}\right)^{3/8}\right]} \tag{24}$$

### 3.6 Estimating Plot and Regional Biomass Density

To test the biomass density equations, we used the results of the MLE fits to calculate the biomass density predicted by Eq.
(7) and Eq. (10). The biomass density predicted by these equations are then compared to the observed biomass density (i.e.
the sum of the mass of all trees in a dataset divided by the area of the plots). This comparison then provides a goodness of fit

measure that is relevant to climate.

We chose to measure the biomass density as a function of size in terms of the total mass per unit area from trees with masses
equal or greater than a given size. The main reason for this is that the forest plot data only sampled trees which have trunk
diameter equal to or greater than 10 cm. Therefore it makes little sense to measure the biomass density below a given size, as
would be the case with a traditional cumulative distribution function. This approach has a second benefit that the mass of a

forest above a given size is much more useful way of easily seeing the contribution of the dominant larger trees to total biomass
(Bastin et al., 2018).

A correction terms is added to Eq. (7) and Eq. (10) to make sure the biomass density correctly evaluates at the upper boundary
(the mass of the largest tree $m_{\max}$). The biomass density of trees equal or greater than $m_{\max}$ should be $m_{\max}/A$, where $A$ is the





total area of plots in the dataset. As when the second term in Eq. (7) is included the biomass density of trees equal or greater than $m_{\max}$ is zero, so we need to add a correction term of $m_{\max}/A$.

$$M_{L \to max} = n_r m_r^{\phi_m} \frac{\exp(x\mu_{m,1} m_r^{1/x})}{\mu_{m,1}(x\mu_{m,1})^x} \left[ \Gamma(x+1, x\mu_{m,1} m_L^{1/x}) - \Gamma(x+1, x\mu_{m,1} m_{\max}^{1/x}) \right] + \frac{m_{\max}}{A} \qquad (25)$$

This Eq. (25) is used for all biomass density estimates where the upper bound of tree size is assumed finite (based on $m_{\max}$), while for the cases where the simplifying assumption of infinite tree size is used then Eq. (7) is used.

## 4 Results

### 4.1 Mass Distribution

When the mass data was estimated from the trunk diameter measurements using the methodology of Feldpausch et al. (2012), it was noticed that the mass size-distribution (for all regions and plots) had a peak, which was not present in the trunk diameter distribution. We found this to be an artefact of the conversion from trunk diameter to mass in a distribution that was by definition truncated already in trunk diameter.

Fig. 2a shows the relationship between trunk diameter and tree mass for the whole dataset, illustrating that for any particular trunk diameter there is a range of tree masses. This variation to mass is is caused by the differences in wood density between species and the variation in height allometry between regions (see Eq. eqn:Height and Table tab:AllomCoefficients). If instead the dataset shown in Fig. 2a is truncated in mass rather than trunk diameter, then the truncation would instead follow the horizontal dotted line and there would be data in the region between that line and the diagonal dotted line. So in effect there is "missing" data for low mass trees, which is a result of the trunk diameter observations having a minimum sampling size (truncation point) and there being a range of tree masses for trees with a given trunk diameter. This hypothesis is further confirmed by increasing the trunk diameter truncation point, as shown in Fig. 2b. As the truncation point is increased the "peak" moves to higher mass.



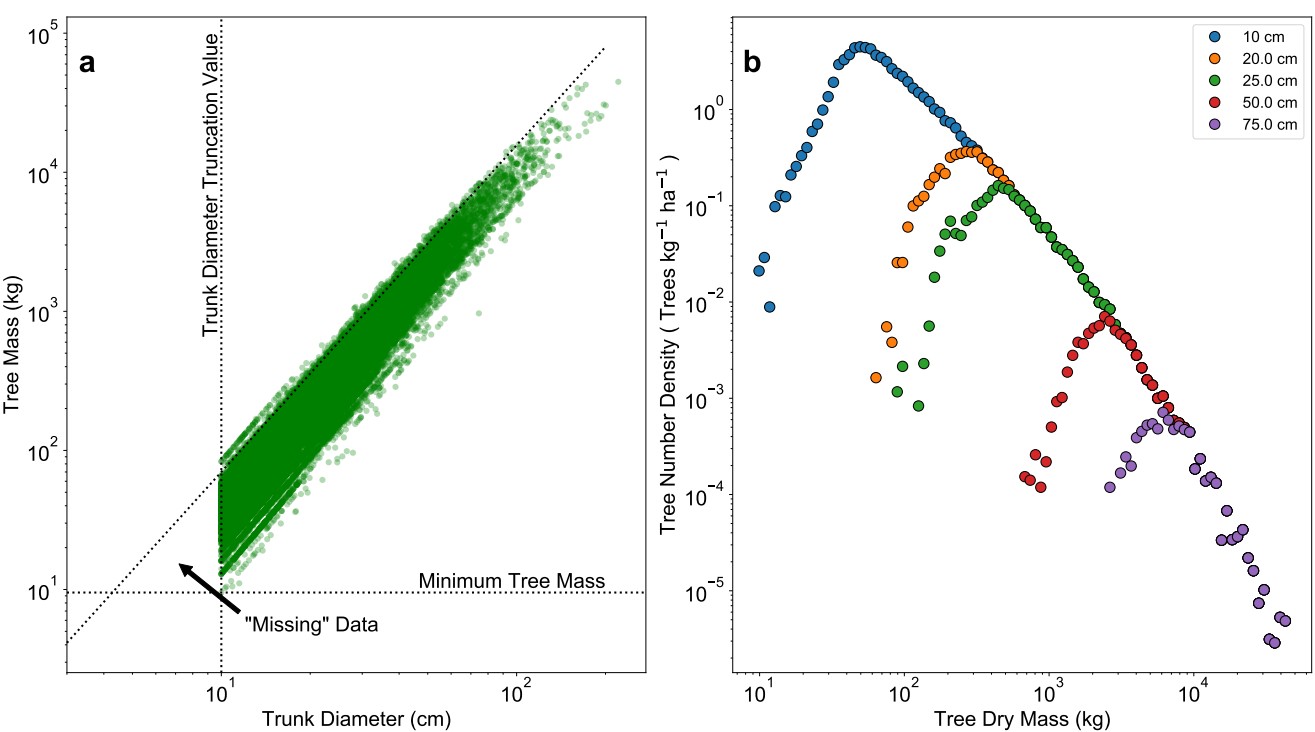

**Figure 2.** The effect of truncating data measured in trunk diameter and then converting to mass using allometry. In **a)**, the mass for each tree in terms of its trunk diameter. If the data had been truncated to mass there would be data in the triangle marked by the intersection of the dotted lines. This truncation effectively leads to missing data in the mass distribution, as seen in **b)**. The mass distribution should constantly decrease with increasing mass but instead rises to a peak then decreases, and is due to incomplete data for the low mass end of the distribution. This peak can be seen to be an artefact of the trunk diameter truncation point. When the trunk diameter truncation point increases the mass distribution peak moves with the truncation point.



## 4.2 Trunk Diameter Results

Fitting the DET-LTWD and MST equations to the trunk diameter size distributions, showed a consistent pattern for all the geographical aggregations of plot data. In all cases, except Guyana shield, the DET-LTWD solutions (both one and two parameter versions) fitted much better than the MST solution (Fig. 3 and see supplementary material Fig. S1 and S2). Guyana Shield

5   region only had four small plots totalling 819 trees which may explain the reason it was hard to distinguish visually the best fitting model. The two parameter DET-LTWD fits gave a fitted value of the growth scaling power $\phi$ between 0.137 and 0.546 (Table 2) and five the twelve regions were within 0.05 of the theoretical value of 1/3 (i.e. $\phi$ in range 0.28-0.38).

**Figure 3.** Fit to the trunk diameter size-distribution for all South American RAINFOR Plots as one large dataset. The blue circles show the binned data and the lines show the fitted distribution for each model.





**Table 2.** Results of fitting the models to the trunk diameter size-distributions for the forest plot data aggregated to regions, countries and the whole continent. Table presents the fitted parameters for each model. $n_L$ is the value of the distribution at the lower truncation point of 10 cm trunk diameter. $\mu_1$ and $\phi$ are the model parameters from Eq. 3 fitted to the data by MLE.

| Region | No.Trees | Area ha | mean D cm | DET 1 Param $\mu_1$ | DET 1 Param $n_L$ (cm ha)$^{-1}$ | DET 2 Param $\mu_1$ | DET 2 Param $\phi$ | DET 2 Param $n_L$ (cm ha)$^{-1}$ | MST $n_L$ (cm ha)$^{-1}$ |
|---|---|---|---|---|---|---|---|---|---|
| All S.America | 63605 | 113.4 | 20.45 | 0.255 | 66.53 | 0.308 | 0.397 | 69.13 | 58.77 |
| Brazil | 12454 | 23.5 | 20.83 | 0.247 | 60.87 | 0.266 | 0.358 | 61.78 | 55.83 |
| Bolivia | 8963 | 16.0 | 20.11 | 0.265 | 68.93 | 0.491 | 0.546 | 78.15 | 59.49 |
| Colombia | 7288 | 13.2 | 19.68 | 0.273 | 69.87 | 0.314 | 0.382 | 71.84 | 58.31 |
| Ecuador | 4949 | 7.8 | 20.37 | 0.257 | 75.82 | 0.330 | 0.419 | 79.79 | 67.37 |
| Peru | 27080 | 44.5 | 20.38 | 0.256 | 72.41 | 0.281 | 0.366 | 73.81 | 63.74 |
| Venezuela | 2871 | 5.3 | 22.55 | 0.217 | 54.95 | 0.204 | 0.313 | 54.20 | 57.71 |
| **Amazonian Allometric Regions** | | | | | | | | | |
| N.Western | 22642 | 37.8 | 20.21 | 0.261 | 72.55 | 0.325 | 0.409 | 75.85 | 63.34 |
| S.Western | 24690 | 42.5 | 20.58 | 0.252 | 67.98 | 0.263 | 0.348 | 68.59 | 60.85 |
| Brazilian Shield | 13412 | 24.5 | 20.10 | 0.264 | 67.21 | 0.399 | 0.476 | 73.07 | 58.18 |
| Guyana Shield | 819 | 1.5 | 22.74 | 0.214 | 54.10 | 0.120 | 0.137 | 47.54 | 61.78 |
| Eastern-Central | 2042 | 4.0 | 22.90 | 0.213 | 50.40 | 0.212 | 0.332 | 50.34 | 53.73 |

In general the one and two parameter DET-LTWD solutions were quite similar in terms of the appearance of the fit on the distribution plots. This finding was confirmed using the Akaike information criterion (AIC) and Bayesian information criterion (BIC) (Table 3). Both the AIC and BIC are a way of determining from several models which represents the data better, with a lower value indicating a better fit. Both criterions are calculated from the log likelihood and number of fitting parameters with a difference of 10 being the threshold where the evidence is considered to be very strongly against the higher scoring model (Kass and Raftery, 1995). BIC penalises a higher number of fitting parameters more than AIC.

It was only possible to distinguish the quality of the fits for four of the twelve geographical aggregations of forest plots. In all four cases (All S.America, Bolivia, Brazilian Shield and N.Western) the two parameter DET-LTWD fit was favoured and for the other eight it was not possible to say that the inclusion of the growth scaling power as a fitting parameter improved the fit.





**Table 3.** Model comparison for fits to trunk diameter size-distributions. Table shows the log Likelihood of each models fit and the corresponding AIC and BIC model comparison criterion. The best model has the lowest AIC or BIC; here the difference is shown to the best model, meaning the best model has a score of 0. Models other than the best are strongly rejected if they have a value greater than 10. Best model and those not rejected are shown in bold.

| Region | log Likelihood | | | Δ AIC | | | Δ BIC | | |
|---|---|---|---|---|---|---|---|---|---|
| | MST | DET 1 Param | DET 2 Param | MST | DET 1 Param | DET 2 Param | MST | DET 1 Param | DET 2 Param |
| All S.America | -218,530 | -211,726 | -211,699 | 13700.0 | 51.6 | **0.0** | 13600.0 | 42.5 | **0.0** |
| Brazil | -43,146 | -41,934 | -41,933 | 2420.0 | **0.0** | **0.404** | 2410.0 | **0.0** | 7.83 |
| Bolivia | -30,243 | -29,433 | -29,389 | 1710.0 | 87.3 | **0.0** | 1690.0 | 80.2 | **0.0** |
| Colombia | -24,577 | -23,715 | -23,714 | 1720.0 | **1.57** | **0.0** | 1720.0 | **0.0** | 5.32 |
| Ecuador | -16,889 | -16,428 | -16,424 | 927.0 | **5.82** | **0.0** | 915.0 | **0.0** | 0.682 |
| Peru | -93,037 | -90,049 | -90,046 | 5980.0 | **3.38** | **0.0** | 5970.0 | **0.0** | 4.83 |
| Venezuela | -10,289 | -10,098 | -10,098 | 379.0 | **0.0** | **1.67** | 373.0 | **0.0** | 7.63 |
| **Amazonian Allometric Regions** | | | | | | | | | |
| N.Western | -77,148 | -74,830 | -74,817 | 4660.0 | 25.9 | **0.0** | 4640.0 | 17.8 | **0.0** |
| S.Western | -85,245 | -82,584 | -82,583 | 5320.0 | **0.0** | **0.883** | 5310.0 | **0.0** | 9.0 |
| Brazilian Shield | -45,391 | -44,107 | -44,077 | 2620.0 | 57.5 | **0.0** | 2610.0 | 50.0 | **0.0** |
| Guyana Shield | -2,901 | -2,898 | -2,895 | **7.41** | **4.76** | **0.0** | **0.0** | **2.06** | **2.01** |
| Eastern-Central | -7,370 | -7,232 | -7,232 | 274.0 | **0.0** | **2.0** | 268.0 | **0.0** | 7.62 |



## 4.3 Trunk Diameter Results for Individual Plots

Fitting the models to the individual forest plots (full results in supplementary material Tables S2 and S3 and Fig. S5 to S13) again resulted in the DET-LTWD models fitting much better than MST. Table 4 shows the results of BIC comparison of the models for the 124 forest plots. In every case, the best model is determined by the lowest BIC value. Inferior models are only

5 considered strongly rejected if their BIC is greater than the best model by 10 or more. This is represented by the columns in the table and shows the one parameter DET-LTWD was the best model by far the most (81 plots). However, in none of those plots was it possible to strongly reject both the other models. The most common result (75 plots) was of the one parameter DET-LTWD being the best model, MST being rejected but the two parameter DET-LTWD also so closely fitting the data it cannot be rejected. The next most common result (17 plots) was the reverse with again MST rejected but the two parameter

DET-LTWD now narrowly better but not sufficient to strongly reject the one parameter DET-LTWD. The MST model was the best model for 15 plots and for 5 of those (ELD_01, ELD_02, RIO_01, RIO_02, TIP_03) the two DET-LTWD models were both strongly rejected. Four of these plots though had very low number of trees, so it would be less likely to be able to pick a model with as much confidence from a distribution of only 100 trees.

**Table 4.** Shows the best and acceptable models for the 124 individual forest plots for trunk diameter. Models are labelled as (M) for MST, (1) for one parameter DET-LTWD and (2) for two parameter DET-LTWD. Columns refer to best fitting model (lowest BIC score). Rows refer to models that are so good a fit compared to the best that they cannot be rejected, as their BIC score is so close to the best model. For example '1M' means the MST and one parameter models are not rejected but two parameter model is rejected based on BIC. Then the columns in this row show how many forest plots have either (1) or (M) model as best fit and the other also fitting closely.

| Comparable | Best Model | | | Total |
|---|---|---|---|---|
| Models | 1 | 2 | M | |
| 1 | 0 | - | - | 0 |
| 2 | - | 8 | - | 8 |
| M | - | - | 5 | 5 |
| 12 | 75 | 17 | - | 92 |
| 1M | 0 | - | 2 | 2 |
| 2M | - | 1 | 1 | 2 |
| 12M | 6 | 2 | 7 | 15 |
| Total | 81 | 28 | 15 | 124 |

Fig. 4 shows the effect of fitting with the two parameter DET-LTWD model. There is to be a clear relationship between $\phi$

and $\mu_1$, as all results follow a curve. The black dotted vertical line shows the $\phi = 1/3$ value expected according to the MST allometry and also assumed in the one parameter DET-LTWD model.





**Figure 4.** Results of the two parameter DET-LTWD MLE fits for trunk diameter data from all 124 individual forest plots. The fitted mortality to growth ratio $\mu_1$ is shown as a function of the fitted growth scaling power $\phi$. The results from the fits to the grouped datasets of the four allometric regions are plotted as the dashed crosses of the corresponding colour. The vertical black line shows the $\phi$ value predicted by MST allometry.



Plotting just the $\phi$ results in a histogram (Fig. 5), reveals an approximate bell-shaped distribution with a peak around the theoretical MST value. The median of the $\phi$ value for the plots is 0.34 (95% confidence interval 0.29-0.40) and the mean is 0.31 (95% confidence interval 0.26-0.36). These values are close to the theoretical value of 1/3, as suggested by the histogram.

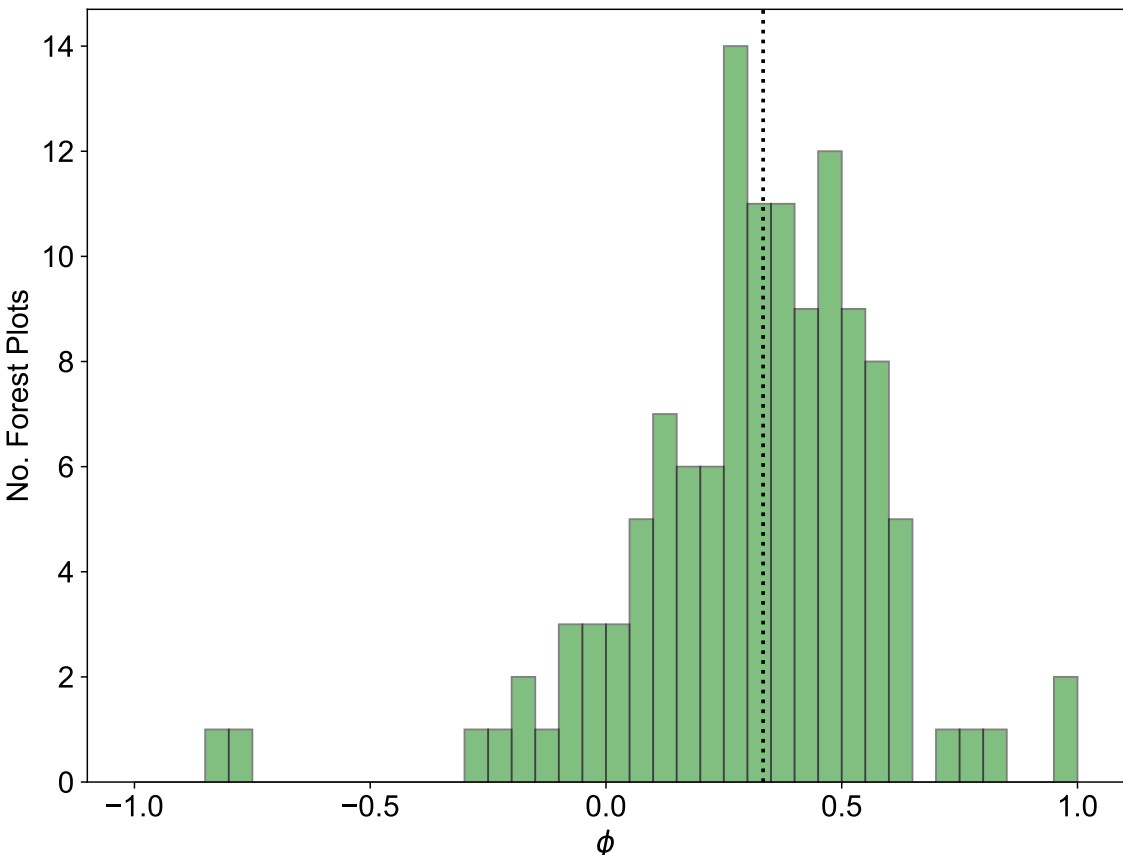

**Figure 5.** Results for the growth scaling power $\phi$ when fitting the two parameter DET-LTWD via MLE for trunk diameter data from all 124 individual forest plots. The vertical black line shows the value $\phi = 1/3$ predicted by MST allometry.





## 4.4 Mass Results

When fitting the DET-LTWD and MST equations to the mass size distributions, there was again a consistent pattern for all the geographical aggregations of plot data. In all cases the DET-LTWD solutions (both one and two parameter versions) fitted much better than the MST solution (Fig. 6 and see supplementary material Fig. S3 and S4).

**Figure 6.** Fit to the mass size-distribution for all South American RAINFOR Plots as one large dataset. The blue circles show the binned data and the lines show the fitted distribution for each model. The peak in the distribution is clearly shown. The fitting is only performed on trees with mass greater than the mass of the peak.

5   The two parameter fits gave a fitted value of the growth scaling power $\phi_m$ between 0.635 and 0.794 (Table 5) which showed that the growth allometry is close to the theoretical value of 0.75 (10 of 12 regions with $\phi_m$ in range 0.7-0.8). The table also


shows the truncation point $m_P$ used for each dataset, and all trees with mass less than this value were excluded. The value of $m_P$ corresponds to the peak in distribution created by the conversion from trunk diameter to mass data. The observed biomass density agrees with the values found previously by Feldpausch et al. (2012), using the same methodology. As this biomass density value is dry mass then it is a reasonable approximation (Chave et al., 2005; Martin and Thomas, 2011) to halve these

5    values to obtain the carbon biomass density, giving a range of 10-15 kg C m$^{-2}$.

**Table 5.** Results of fitting the models to the mass size-distributions for the forest plot data aggregated to regions, countries and the whole continent. Shown are the fitted parameters for each model. $m_P$ refers to the point which all data with smaller mass was excluded to remove the allometry conversion artefact. $n_P$ is the value of the distribution at tree mass of $m_P$. Biomass is the tree dry mass density of all trees above $m_P$.

| | | | | | DET 1 Param | | DET 2 Param | | | MST |
|---|---|---|---|---|---|---|---|---|---|---|
| Region | No.Trees | Area ha | $m_P$ kg | Biomass kg m$^{-2}$ | $\mu_{m,1}$ | $n_p$ (kg ha)$^{-1}$ | $\mu_{m,1}$ | $\phi_m$ (kg ha)$^{-1}$ | $n_p$ (kg ha)$^{-1}$ | $n_p$ (kg ha)$^{-1}$ |
| All S.America | 56702 | 113.36 | 46.4 | 22.2 | 0.198 | 5.58 | 0.189 | 0.741 | 5.51 | 4.38 |
| Brazil | 10719 | 23.48 | 45.6 | 22.1 | 0.193 | 5.02 | 0.212 | 0.768 | 5.15 | 4.11 |
| Bolivia | 7892 | 16.00 | 40.6 | 21.7 | 0.199 | 6.12 | 0.225 | 0.773 | 6.33 | 4.92 |
| Colombia | 6167 | 13.21 | 55.5 | 19.0 | 0.216 | 4.95 | 0.188 | 0.724 | 4.79 | 3.45 |
| Ecuador | 4159 | 7.80 | 54.5 | 23.1 | 0.208 | 5.53 | 0.240 | 0.777 | 5.73 | 4.12 |
| Peru | 22414 | 44.50 | 57.3 | 23.5 | 0.204 | 4.93 | 0.194 | 0.741 | 4.87 | 3.59 |
| Venezuela | 2437 | 5.27 | 64.9 | 30.6 | 0.167 | 3.38 | 0.115 | 0.684 | 3.06 | 2.99 |
| **Amazonian Allometric Regions** | | | | | | | | | | |
| N.Western | 20016 | 37.78 | 51.5 | 22.9 | 0.203 | 5.59 | 0.187 | 0.735 | 5.47 | 4.20 |
| S.Western | 20375 | 42.50 | 57.3 | 22.6 | 0.205 | 4.72 | 0.204 | 0.749 | 4.71 | 3.42 |
| Brazilian Shield | 11460 | 24.48 | 40.6 | 20.2 | 0.204 | 5.93 | 0.249 | 0.789 | 6.27 | 4.67 |
| Guyana Shield | 765 | 1.50 | 59.1 | 38.8 | 0.148 | 3.54 | 0.083 | 0.648 | 3.00 | 3.79 |
| Eastern-Central | 1773 | 4.00 | 51.8 | 32.7 | 0.157 | 3.61 | 0.147 | 0.737 | 3.54 | 3.53 |

As with the trunk diameter, fits for the two DET-LTWD solutions were, in general, quite similar in terms of the appearance on the mass distribution plots. Again the AIC and BIC fitting metrics were barely able to distinguish which DET-LTWD model best fit the data (Table 6). Nine of the geographical aggregations (All S.America, Brazil, Bolivia, Colombia, Ecuador, Peru, N.Western, Guyana Shield and Eastern Central) all could not distinguish the DET-LTWD fits in either AIC or BIC. For

10   Venezuela AIC indicated that the two parameter fit may be slightly better but BIC was not able to show any difference. The S.Western allometric region was the only one showing the one parameter fit as being better but only for BIC. The only region to have both AIC and BIC favouring one of the fits was the Brazilian Shield region, where both AIC and BIC favoured the two parameter fit.





**Table 6.** Model comparison for fits to mass size-distributions. Table shows the log Likelihood of each models fit and the corresponding AIC and BIC model comparison criterion. The best model has the lowest AIC or BIC; here the difference is shown to the best model, meaning the best model has a score of 0. Models other than the best are strongly rejected if they have a value greater than 10. Best model and those not rejected are shown in bold.

| Region | log Likelihood | | | Δ AIC | | | Δ BIC | | |
|---|---|---|---|---|---|---|---|---|---|
| | MST | DET 1 Param | DET 2 Param | MST | DET 1 Param | DET 2 Param | MST | DET 1 Param | DET 2 Param |
| All S.America | -378,596 | -371,541 | -371,538 | 14100.0 | **3.68** | **0.0** | 14100.0 | **0.0** | **5.38** |
| Brazil | -71,653 | -70,609 | -70,607 | 2090.0 | **2.54** | **0.0** | 2080.0 | **0.0** | **4.89** |
| Bolivia | -51,899 | -51,009 | -51,006 | 1780.0 | **3.58** | **0.0** | 1770.0 | **0.0** | **3.52** |
| Colombia | -41,118 | -40,122 | -40,119 | 1990.0 | **2.21** | **0.0** | 1980.0 | **0.0** | **4.69** |
| Ecuador | -27,700 | -27,241 | -27,240 | 917.0 | **1.24** | **0.0** | 909.0 | **0.0** | **5.26** |
| Peru | -151,615 | -148,379 | -148,378 | 6470.0 | **0.0** | **0.004** | 6460.0 | **0.0** | **8.21** |
| Venezuela | -17,382 | -17,204 | -17,198 | 364.0 | 10.8 | **0.0** | 352.0 | **4.83** | **0.0** |
| **Amazonian Allometric Regions** | | | | | | | | | |
| N.Western | -134,204 | -131,530 | -131,528 | 5350.0 | **3.13** | **0.0** | 5340.0 | **0.0** | **4.89** |
| S.Western | -137,602 | -134,629 | -134,629 | 5940.0 | **0.0** | **1.97** | 5940.0 | **0.0** | 10.1 |
| Brazilian Shield | -74,940 | -73,604 | -73,592 | 2690.0 | 20.8 | **0.0** | 2680.0 | 13.3 | **0.0** |
| Guyana Shield | -5,545 | -5,547 | -5,541 | **3.39** | 8.46 | **0.0** | **0.0** | 9.78 | **6.03** |
| Eastern-Central | -12,577 | -12,487 | -12,487 | 178.0 | **0.0** | **1.58** | 172.0 | **0.0** | **7.2** |





## 4.5 Mass Results for Individual Plots

Fitting the models to the individual forest plots (full results in supplementary material Tables S3 and S4 and Fig. S14 to S22) again resulted in the DET-LTWD models fitting much better than MST. Table 7 shows the results of BIC comparison of the models for the 124 forest plots. In every case, the best model is determined by the lowest BIC value. Inferior models are only

considered strongly rejected if their BIC is greater than the best model by 10 or more. This is represented by the columns in the table and shows the one parameter DET-LTWD was the best model by far the most (80 plots). However, in none of those plots was it possible to strongly reject both the other models. The most common result (74 plots) was of the one parameter DET-LTWD being the best model, MST being rejected but the two parameter DET-LTWD also so closely fitting the data it cannot be rejected. The next most common result (14 plots) was the reverse with again MST rejected but the two parameter

DET-LTWD narrowly better but not sufficient to strongly reject the one parameter DET-LTWD. The MST model was the best model for 15 plots and for 5 of those (ELD_01, ELD_02, RIO_01, SUC_03, TIP_03) the two DET-LTWD models were both strongly rejected. Three of these plots though had very low number of trees so it would be less expected to be able to accurately pick a model from a distribution of only  100 trees.

**Table 7.** Shows the best and acceptable models for the 124 individual forest plots for mass. Models are labelled as (M) for MST, (1) for one parameter DET-LTWD and (2) for two parameter DET-LTWD. Columns refer to best fitting model (lowest BIC score). Rows refer to models that are so good a fit compared to the best that they cannot be rejected, as their BIC score is so close to the best model. For example '1M' means the MST and one parameter models are not rejected but two parameter model is rejected based on BIC. Then the columns in this row show how many forest plots have either (1) or (M) model as best fit and the other also fitting closely.

| Comparable Models | Best Model | | | Total |
|---|---|---|---|---|
| | 1 | 2 | M | |
| 1 | 0 | - | - | 0 |
| 2 | - | 11 | - | 11 |
| M | - | - | 5 | 5 |
| 12 | 74 | 14 | - | 88 |
| 1M | 0 | - | 2 | 2 |
| 2M | - | 3 | 2 | 5 |
| 12M | 6 | 1 | 6 | 13 |
| Total | 80 | 29 | 15 | 124 |

Fig. 7 shows the effect of fitting with the two parameter DET-LTWD model. There is to be a clear relationship between $\phi_m$

and $\mu_{m,1}$, as all results follow a curve. The black dotted vertical line shows the $\phi_m = 0.75$ value expected according to the MST allometry and also assumed in the one parameter DET-LTWD model.



**Figure 7.** Results of the two parameter DET-LTWD MLE fits for mass data from all 124 individual forest plots. The fitted mortality to growth ratio $\mu_{m,1}$ is shown as a function of the fitted growth scaling power $\phi_m$. The results from the fits to the grouped datasets of the four allometric regions are plotted as the dashed crosses of corresponding colour. The vertical black line shows the $\phi_m$ value predicted by MST allometry.





Plotting just the $\phi$ results in a histogram (Fig. 8), reveals an approximate bell-shaped distribution with a peak around the theoretical MST value. The median of the $\phi_m$ value for the plots is 0.72 (95% confidence interval 0.71-0.75) and the mean is 0.71 (95% confidence interval 0.69-0.73). These values are close to the theoretical value of 0.75, as suggested by the histogram.

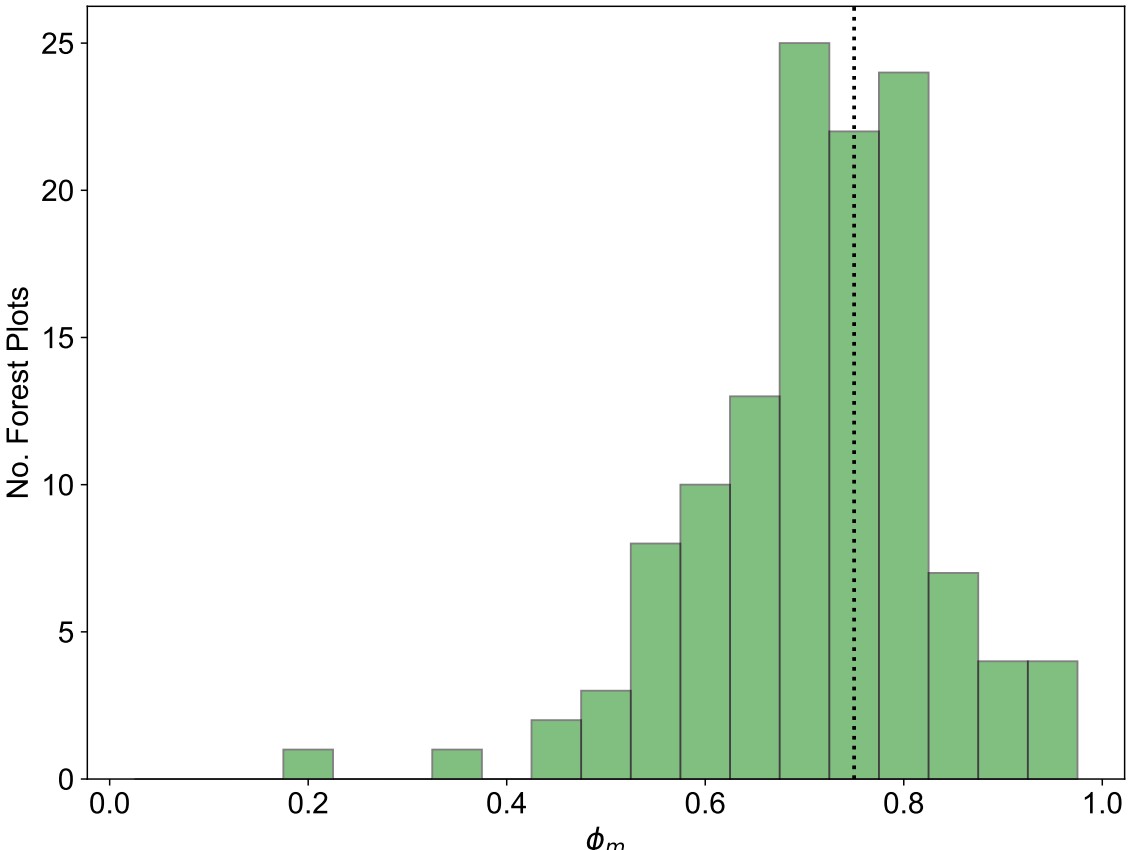

**Figure 8.** Results for the growth scaling power $\phi_m$ when fitting the two parameter DET-LTWD via MLE for mass data from all 124 individual forest plots. The vertical black line shows the value $\phi_m = 0.75$ predicted by MST allometry.



### 4.6 Biomass Results

The biomass density equations Eq. (6), Eq. (7) and Eq. (10) were tested against the observed biomass density (summed tree mass data), as can be seen in Table 8. The biomass density equation parameters were obtained from the fits in Table 5. For the DET-LTWD solutions the biomass density was calculated for both the cases where the upper bound was infinity and the maximum tree mass in the dataset. For each of those cases, the one and two parameter DET-LTWD solutions were calculated.

It is apparent that the MST biomass density equation is inferior to DET-LTWD derived biomass density equation from the DET theory. For all aggregations the biomass density was overestimated by MST, and in many cases by a considerable margin. The comparison of the different DET-LTWD biomass density equations was found to favour the two parameter fit using the finite upper bound (6 regions out of 12). Four areas had better estimates with the two parameter fit using the infinite upper bound (All S.America, Bolivia, Peru and Guyana Shield).

Interestingly, two regions (S.Western and Ecuador) had a worse fit for two parameter DET-LTWD; this appears to be due to MLE fitting favouring getting the fit correct for the more numerous smaller trees rather than the larger rarer trees. When the reverse cumulative biomass density, defined as biomass density of all trees above a given tree mass, is plotted (see supplementary material) the error comes from the shape of the tail of the distribution, which is much flatter than theory. This could be due to it being a region with a smaller number of trees (4188) or could be due to higher mortality for large trees in this region.





**Table 8.** Model Biomass Comparison. Table shows the percentage difference between each model of the biomass density predicted by the parameters obtained from fitting the mass distribution using MLE, and the observed mass in the dataset. This comparison is only for data where the tree mass is greater than the peak in the mass distribution $m_P$. Bold indicates the model that is the closest fit to the observed value.

| | $m_P$ | Observed | % Difference to Observed Biomass Density | | | | |
| --- | --- | --- | --- | --- | --- | --- | --- |
| | | Biomass | LTWD $m_P$ to $\infty$ | | LTWD $m_P$ to $m_{\max}$ | | MST |
| | kg | kg m$^{-2}$ | DET 1 Param | DET 2 Param | DET 1 Param | DET 2 Param | |
| All S.America | 46.4 | 22.2 | 1.62% | **-0.09%** | 0.67% | -0.77% | 389.9% |
| Brazil | 45.6 | 22.1 | -1.67% | 2.21% | -3.79% | **-1.34%** | 253.5% |
| Bolivia | 40.6 | 21.7 | -4.62% | **0.30%** | -4.50% | -0.68% | 355.7% |
| Colombia | 55.5 | 19.0 | 2.01% | -1.69% | 2.88% | **-0.41%** | 439.6% |
| Ecuador | 54.5 | 23.1 | 1.97% | 7.10% | **-0.54%** | 2.10% | 226.5% |
| Peru | 57.3 | 23.5 | 1.17% | **-0.31%** | 0.67% | -0.58% | 407.4% |
| Venezuela | 64.9 | 30.6 | 16.09% | 2.70% | 8.45% | **1.62%** | 170.6% |
| **Amazonian Allometric Regions** | | | | | | | |
| N.Western | 51.5 | 22.9 | 5.13% | 2.43% | 4.37% | **2.12%** | 394.7% |
| S.Western | 57.3 | 22.6 | **-1.12%** | -1.32% | -1.51% | -1.68% | 402.4% |
| Brazilian Shield | 40.6 | 20.2 | -7.04% | 1.33% | -7.10% | **-0.72%** | 364.4% |
| Guyana Shield | 59.1 | 38.8 | 29.40% | **4.58%** | -8.53% | -8.25% | 28.7% |
| Eastern-Central | 51.8 | 32.7 | 8.76% | 5.30% | 1.72% | **0.25%** | 143.7% |




## 4.7 Biomass Results for Individual Plots

To look deeper at the relationship between model choice and predicted biomass density the analysis was repeated for the individual forest plots. In Fig. 9, the results of the biomass density predicted by the models is shown as a function of the actual observed biomass density. It can be observed that correcting for the largest tree size in each plot is much better than assuming an infinite maximum tree size and that the one parameter model performs less well for finite maximum tree size case.

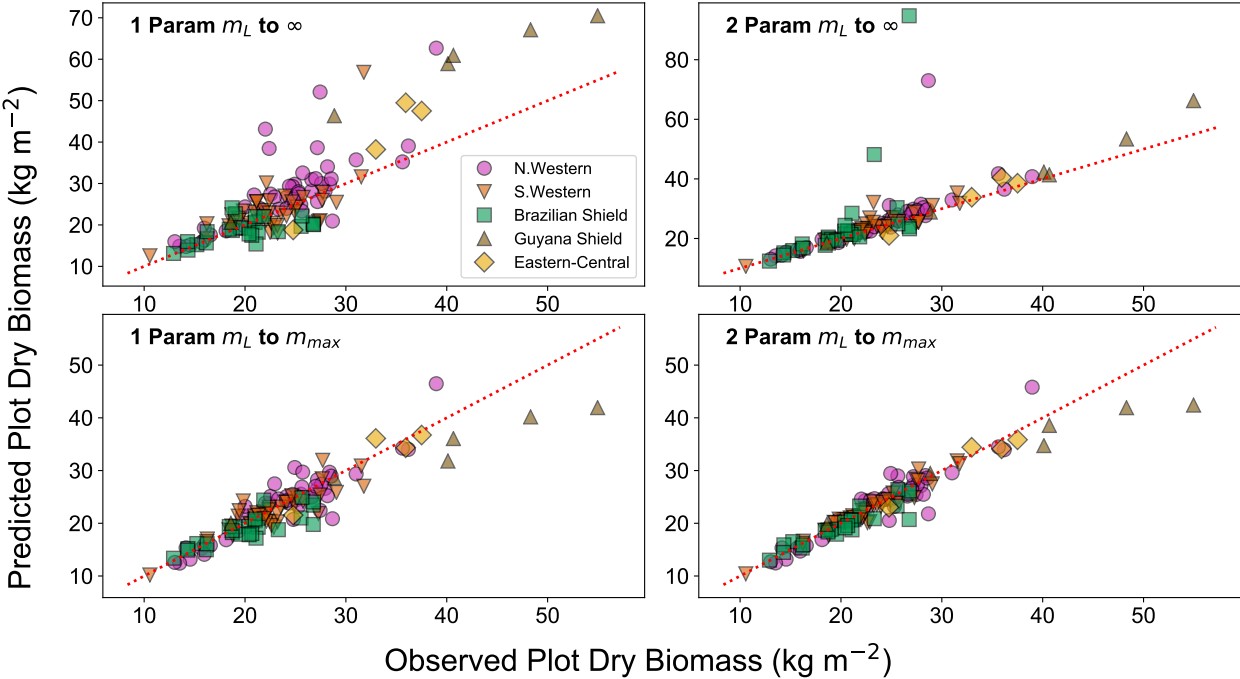

**Figure 9.** Comparison of the biomass density prediction based of the size-distribution fits to the mass data, to the observed biomass density in each of the 124 forest plots. Results are plotted for both the one and two parameter fits and for both the assumption of infinite and finite maximum tree size. The finite tree size case is limited to the largest tree mass $m_{\max}$ in each forest plot.

This finding is supported by looking at the relative root mean squared error (root mean squared error divided by observed biomass density) for each model, as shown in Table 9.

For the small individual forest plots, finite maximum tree size has a larger effect on accuracy than using the two parameter DET-LTWD over the one parameter version.





**Table 9.** The relative root mean squared error (RMSE) of the biomass density prediction of the 124 forest plots using the parameters fitted via MLE to the mass size-distribution. The table compares the results from the different DET-LTWD models and the MST model. The range column indicates the integration limits of the biomass density calculation. The DET-LTWD model assumes no maximum size and by default integrates out to infinity. This can be corrected in terms of the largest tree mass $m_{\max}$ in the dataset.

| Model | Range | Relative RMSE |
|---|---|---|
| 1 Parameter DET-LTWD | $m_P$ to $\infty$ | 0.236 |
| 2 Parameter DET-LTWD | $m_P$ to $\infty$ | 0.295 |
| 1 Parameter DET-LTWD | $m_P$ to $m_{\max}$ | 0.098 |
| 2 Parameter DET-LTWD | $m_P$ to $m_{\max}$ | 0.069 |
| MST | $m_P$ to $m_{\max}$ | 1.387 |





## 5 Discussion

In this paper we investigated which of three demographic models (MST and two forms of DET) could be best used to model the size-distributions and total biomass density of the Amazon forest. This analysis was done for both trunk diameter measurements and for those same measurements converted allometrically (Feldpausch et al., 2012) to mass. This conversion introduces a peak

in the mass distribution that is purely an artefact of the conversion. The peak is due to the variation in mass of trees of a given trunk diameter, due to height and wood density variation leading to some small mass trees being in effect "missing" from the mass distribution. If the diameter to mass relationship was purely one-to-one, then the artefact peak would not occur. This has implications for anyone using mass size-distributions converted from trunk diameter data. Our solution is to only to fit to trees with mass greater than the mass distribution peak.

Each of the three models of size distribution could predict total biomass density by the integration of the analytical form of mass distribution. One interesting implication of the resulting equations is that the mortality and growth only ever appear in the form of the ratio $\mu_1$ and never independently. This suggests that the ratio of mortality to growth determines the equilibrium state of a forest, and that the actual magnitude of the individual mortality and growth terms only determines the transient effects away from a steady state.

Of the three models, MST is rarely a good fit at plot, regional or continental level for either trunk diameter or mass distributions and significantly overestimates total biomass density. The MST was only the undisputed best model in 4% of forest plots and only for 20% was the model a good enough fit not to be rejected. For the aggregated datasets (continent, regions and countries) only the Guyana region was the MST model not rejected, and this region only has a small number of trees, which leads to higher uncertainty in the fit.

The DET model assumed power-law growth with size and size independent mortality, which gives a left truncated Weibull distribution (LTWD). DET-LTWD was tested both when its growth allometry was constrained to the MST growth allometry (West et al., 2009; Niklas and Spatz, 2004) and when the growth allometry parameter was allowed to vary as a fitting parameter (one and two parameter fitting respectively). This comparison allows the MST allometry to be tested independently of the space-filling assumptions of MST. If the MST allometry is true, then the two DET-LTWD models should be in agreement

as the unconstrained version of the model can still select the MST allometry if it is the best fit. The BIC criterion for model selection showed that the MST constrained DET-LTWD was in contention for 89% of forest plots, for five of the six countries and for three of the five regions, and for mass distribution of the whole continent. The unconstrained DET-LTWD was only clearly the best model for the trunk diameter distribution of the whole continent, Bolivia and the Brazilian Shield region.

For the unconstrained DET-LTWD model, the growth scaling power $\phi$ did vary between plots and regions but the me-

30 dian/mean values and the continental result all fell very close to the values predicted by the MST allometry. For the aggregated datasets, the fitted $\phi$ values for the mass distributions were closer to MST than for the trunk diameter distributions. This finding suggests that the MST allometry represents an underlying reality for the scaling of trees that is modified by local conditions. This idea was first raised by Price et al. (2007) and later disputed by Coomes and Allen (2009). It was further suggested (Coomes et al., 2011) that light competition should modify the growth function. This would mean that for trunk diameter the





growth scaling power would vary with size and be greater than the predicted MST value of 1/3. For our regional fits the fitted power, while close to the theoretical value of 1/3, it was larger in most cases but for the individual forest plots the value was very close to MST with no clear bias.

We find the $\phi$ values for both mass and trunk diameter also have a well defined relationship to the fitted mortality : growth
ratio $\mu_1$. This relationship does not appear to be a fitting artefact, as if artificial data is generated with $\mu_1$ and $\phi$ values off the observed curve the fitting process correctly fits it to the generated values. This relationship suggests an interesting but as yet unknown property of the Amazon forests.

When considering how well the models predicted total biomass density from the fitted size-distribution, the biggest source of error at the plot scale is the model assumption of infinite maximum tree size. However, this can be easily corrected for and
allows the MST constrained DET-LTWD to estimate biomass density with relative root mean square error of 10% over the 124 forest plots and unconstrained DET within 6%. Conversely, the MST model consistently overestimated the biomass density, often by a considerable margin. The regional scale, where there were more trees, showed much better prediction of the biomass density and the unconstrained DET-LTWD with finite upper bound had the smallest error in biomass density.

In terms of what this means, again we reject the MST model as a good model of forest size-distributions. This rejection
is consistent with the recent study by Zhou and Lin (2018) that showed there was a fundamental flaw in the derivation of MST. They had noticed that the MST model failed to account for the effect of size-dependent growth rate on how fast a tree transitions through a given size class. This observation explains that the assumptions of MST of the size distribution scaling $D^{-2}$ is inconsistent with the assumption of individual tree resource use scaling as $D^2$. Here, we have confirmed the $D^{-2}$ size-distribution model should be rejected for South American tropical forests. Furthermore, for most plots also we can reject
a general power law distribution also as the distributions observed are rarely linear when plotted in log-log space.

For large scale modelling, such as DGVMs, the extra parameter of the unconstrained DET-LTWD performs better, particularly for biomass density. Despite this, the constrained version still does quite well, and for some regions can actually be better. So, the DET-LTWD model constrained by MST allometry has the benefits of simplicity but at a small cost in accuracy. At the smaller scale of individual forest plots, the variation in fitted growth scaling allometry is greater. At these smaller scales, the
extra allometry parameter in fitting is much more important but even then still allows biomass density modelling with 10% mean error. So, including the extra fitting parameter is a trade-off between simplicity and accuracy and model choice will depend on the needs of the particular application.

## 5.1 Closed Form of DET Solutions for Applications in DGVMs

As Demographic Equilbrium Theory (DET) provides such a good description of the mass-distributions observed in Amazonia,
this suggests that dynamic forest demography (as given by Eq. 1) could form the basis of a next generation Dynamic Global Vegetation Model (DGVM) to be used with an Earth System Model (ESM). For such DGVM applications, the DET model solutions need to be extended by deriving a closed form that includes seedling recruitment. We do this by partitioning a fixed fraction $\alpha$ of the net assimilate $P$ (net primary productivity remaining after accounting for litterfall) of each tree to seedling production, with the remainder of $P$ going into the growth of the tree itself.





This allows a solution to be derived (see Appendix) for the total equilibrium fractional coverage $\nu$ (fraction of ground area covered by tree crowns) that is only a function of three parameters, under the assumption that there is negligible overlap of the crowns in the canopy (Fig. 10).

$$\nu = 1 - \left(\frac{1-\alpha}{\alpha}\right)\frac{\mu_s(x\mu_s)^{x-1}}{\exp(x\mu_s)\Gamma(x,x\mu_s)} \tag{26}$$

5    where $\mu_s = \mu_1 m_s^{(1-\phi)}$ is the mortality to growth ratio at the seedling size $m_s$.

Further solutions can then be derived (see Appendix) for total forest properties such as biomass density $M$ (Fig. 10) and in a way that eliminates dependent terms such as $n_s$ (number density per size class at seedling size). To do this we assume the crown area of the trees in the forest scales as a power law with scaling power $\phi_a$ and seedlings have crown area $a_s$, mass $m_s$ and growth $g_s$.

$$M = \frac{m_s}{a_s}\frac{(x\mu_s)^{x(\phi_a-1)}\Gamma(x+1,x\mu_s)}{\Gamma(\phi_a x+1,x\mu_s)}\nu \tag{27}$$

This means we have simple equilibrium solutions that are a function of known parameters of each plant functional type (PFT) and of parameters that can be provided by the land-surface scheme of an ESM. Equilibrium solutions can also be obtained for a discrete set of size classes, allowing the demographic profile to be initialised in an equilibrium state consistent with observed mass-distributions and forest area coverage. Once such a model is initialised it can be run dynamically to simulate transient states based-on inputs of additional mortality due to disturbance processes and time-varying net assimilate from a driving land-surface scheme. (Argles et al.). We are optimistic that this will allow more realistic simulation of the response of forests to changes in climate and atmospheric carbon dioxide, including possible forest die-back due to increasing temperatures, drought and forest fires.





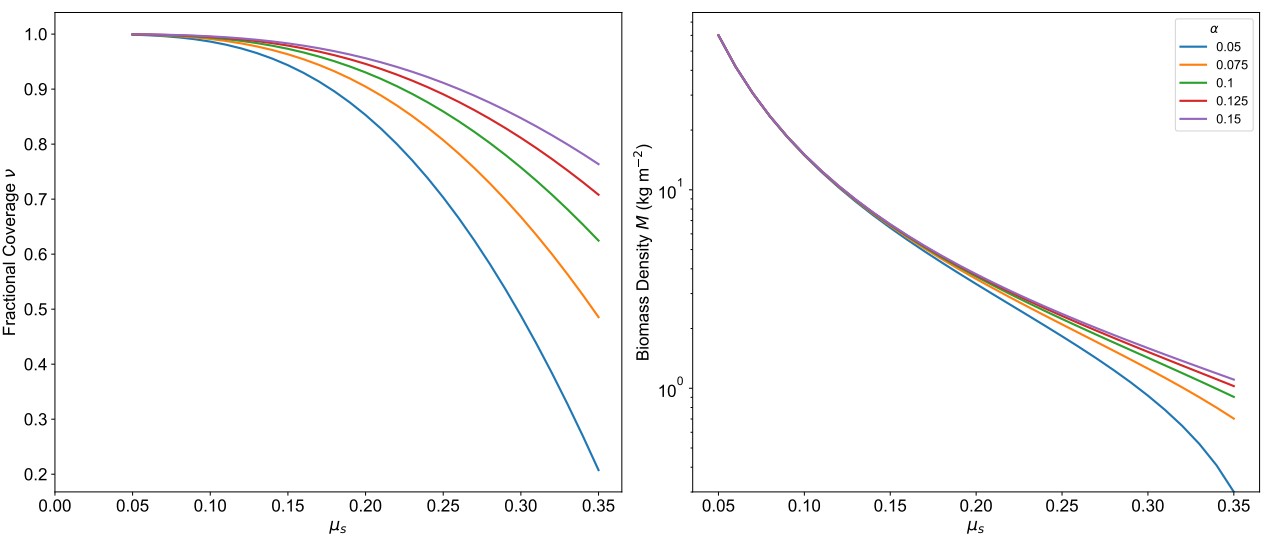

**Figure 10.** Shows the dependence of the total forest fractional coverage $\nu$ and total biomass density $M$ on the mortality to growth ratio of seedlings $\mu_s$ for the closed form solutions where the rate of seedling production is a fixed fraction $\alpha$ of tree productivity. These plots assume $\phi_m = 0.75$, and seedlings with mass $m_s$ of 10g and that a 1 kg tree would have a crown area of 0.5 m$^2$.





# 6   Conclusions

This study demonstrates that demographic equilibrium theory (DET) is able to fit measured tree size-distributions in Amazonian forests. DET was found to outperform Metabolic Scaling-theory (MST, West et al. (2009)), even when using just a single fitting parameter ($\mu_1$ – the ratio of mortality to growth), and assuming the MST allometric relation between tree growth-rate and tree-mass ($g \propto m^{3/4}$, Niklas and Spatz (2004)). When the exponent relating growth-rate to tree-mass $\phi$ was also allowed to vary by site, DET produced an even better fit to the observed tree-mass distribution, and also an intriguing (but as yet unexplained) relationship between $\phi$ and $\mu_1$ across sites. Furthermore, equations derived from DET, predicting the biomass density in terms of $\phi$ and $\mu_1$ are a very climate relevant measure of goodness of fit for these models. The success of DET in explaining tree-size distributions across both North America (Moore et al., 2018) and South America (this study) indicates that relatively simple and robust ecosystem demography is a good basis for a next-generation dynamic global vegetation model (Argles et al., in preparation).

*Code availability.*   Code is available on reasonable request to the corresponding author.

## Appendix A

The closed form model (see Section 5.1) makes the following assumptions: -

- All trees produce seeds and there is no minimum reproductive size threshold.

- All seeds have mass $m_s$.

- Seeds reaching the ground either die or are recruited as a sapling, no seed pool is modelled

- All trees have a crown area that follows a power law with tree mass, and that the crown area is a big leaf and totally opaque with no light allowed through.

- Trees have the minimum overlap in crowns possible (perfect plasticity assumption).

- The proportion of seeds recruited $\omega$ is determined by shading. This proportion $\omega = 1 - \upsilon$, where $\upsilon$ is the fractional coverage of crowns of all trees with mass greater than the seed mass $m_s$.

- A proportion of productivity $\alpha$ is assumed to go into seed production.

The total growth per hectare $G$, for all trees larger than $m_s$, is

$$G = (1 - \alpha)P \tag{A1}$$





where $P$ is the Net Productivity after losses from respiration and litter are accounted for. The amount of mass going into seeds $P_S$ is then defined as

$$P_S = \alpha P \tag{A2}$$

Therefore

$$P_S = \frac{\alpha}{1-\alpha} G \tag{A3}$$

The DET solution describes the size distribution, for tree mass growth rate $g(m) = g_s(m/m_s)^\phi$

$$n(m) = n_s \left(\frac{m}{m_s}\right)^{-\phi} \exp\left\{ \frac{\mu_s}{1-\phi}\left[1 - \left(\frac{m}{m_s}\right)^{1-\phi}\right]\right\}, \quad \phi \neq 1 \tag{A4}$$

The mortality growth ratio $\mu_s$ is a function of $\alpha$ as the individual tree growth $g_s$ is a fraction $(1-\alpha)$ of the tree productivity $p(m) = p_s(m/m_s)^\phi$.

$$\mu_s = \frac{\gamma m_s}{g_s} = \frac{\gamma m_s^{(1-\phi)}}{g_1} = \mu_1 m_s^{(1-\phi)} = \frac{\gamma m_s}{p_s(1-\alpha)} \tag{A5}$$

The equation for the total mass growth from trees with mass from $m_s$ to $\infty$ is

$$G = \int\limits_{m_s}^{\infty} n(m)g(m)dm = g_s N \frac{\exp(x\mu_s)}{(x\mu_s)^{x-1}}\Gamma(x, x\mu_s) \tag{A6}$$

While the tree / stem density $N$ is

$$N = \int\limits_{m_s}^{\infty} n(m)dm = \frac{n_s m_s}{\mu_s} = \frac{n_s g_s}{\gamma} \tag{A7}$$

The boundary condition defines the seed recruitment rate $n_s g_s$ (in units saplings yr$^{-1}$ ha$^{-1}$)

$$n_s g_s = \frac{P_S}{m_s}(1-\upsilon_s) = \frac{\alpha}{1-\alpha}\frac{G}{m_s}(1-\upsilon) \tag{A8}$$

By substituting equation A6 into equation A8 and then rearranging in terms of $\upsilon_s$, and eliminating $N$ using equation A7, a solution for $\upsilon$ can be obtained that only depends on known variables $x(\phi)$, $\alpha$ and $\mu_s$.

$$\upsilon = 1 - \left(\frac{1-\alpha}{\alpha}\right)\frac{\mu_s(x\mu_s)^{x-1}}{\exp(x\mu_s)\Gamma(x, x\mu_s)} \tag{A9}$$





$\upsilon$ is also defined by integrating the crown area weighted by $n(m)$ between $m_s$ and $\infty$

$$\upsilon = \int_{m_s}^{\infty} a(m)n(m)dm = \frac{N\,a_s\,\exp(x\mu_s)}{(x\mu_s)^{\theta_a x}}\Gamma(\theta_a x + 1, x\mu_s) \tag{A10}$$

where the individual tree crown area $a(m) = a_s m^{\phi_a}$.

Rearranging equation A10 gives another equation for $N$

$$N = \frac{1}{a_s}\frac{(x\mu_s)^{\theta_a x}}{\exp(x\mu_s)}\frac{1}{\Gamma(\phi_a x + 1, x\mu_s)}\nu \tag{A11}$$

This result can then be combined with the equations for $G$ (equation A6) and $M$ (equation A12) to obtain results independent of $N$ or $n_s$.

Firstly, the equation for total mass $M$ is

$$M = m_s N\frac{\exp(x\mu_s)}{(x\mu_s)^x}\Gamma(x + 1, x\mu_s) \tag{A12}$$

10 The $N$ independent versions are then

$$M = \frac{m_s\,(x\mu_s)^{x(\theta_a - 1)}}{a_s}\frac{\Gamma(x + 1, x\mu_s)}{\Gamma(\phi_a x + 1, x\mu_s)}\nu \tag{A13}$$

$$G = \frac{g_s\,(x\mu_s)^{x(\theta_a - 1)+1}}{a_s}\frac{\Gamma(x, x\mu_s)}{\Gamma(\phi_a x + 1, x\mu_s)}\nu \tag{A14}$$

For the MST allometry then $x = 4$ and $\theta_a = 1/2$

$$\upsilon = 1 - \left(\frac{1 - \alpha}{\alpha}\right)\frac{\mu_s(4\mu_s)^3}{\exp(4\mu_s)\Gamma(4, 4\mu_s)} \tag{A15}$$

15 $$N = \frac{1}{a_s}\frac{(4\mu_s)^2}{\exp(4\mu_s)}\frac{1}{\Gamma(3, 4\mu_s)}\nu \tag{A16}$$

$$M = \frac{m_s}{a_s}\frac{1}{(4\mu_s)^2}\frac{\Gamma(5, 4\mu_s)}{\Gamma(3, 4\mu_s)}\nu \tag{A17}$$

$$G = \frac{g_s}{a_s}\frac{1}{(4\mu_s)}\frac{\Gamma(4, 4\mu_s)}{\Gamma(3, 4\mu_s)}\nu \tag{A18}$$



For these particular MST allometry values the Gamma functions can be represented by a finite series as when $a$ is an integer in $\Gamma(a,z)$ then

$$\Gamma(a,z) = (a-1)!e^{-z}\sum_{k=0}^{a-1}\frac{z^k}{k!} \tag{A19}$$

$$\upsilon = 1 - \left(\frac{1-\alpha}{\alpha}\right)\frac{\mu_s}{\left(1+\frac{3}{4\mu_s}+\frac{3}{8\mu_s^2}+\frac{3}{32\mu_s^3}\right)} \tag{A20}$$

$$N = \frac{\nu}{a_s}\left(\frac{1}{1+\frac{1}{2\mu_s}+\frac{1}{8\mu_s^2}}\right) \tag{A21}$$

$$G = \frac{\nu g_s}{a_s}\left(\frac{1+\frac{3}{4\mu_s}+\frac{3}{8\mu_s^2}+\frac{3}{32\mu_s^3}}{1+\frac{1}{2\mu_s}+\frac{1}{8\mu_s^2}}\right) \tag{A22}$$

$$M = \frac{\nu m_s}{a_s}\left(\frac{1+\frac{1}{\mu_s}+\frac{3}{4\mu_s^2}+\frac{3}{8\mu_s^3}+\frac{3}{32\mu_s^4}}{1+\frac{1}{2\mu_s}+\frac{1}{8\mu_s^2}}\right) \tag{A23}$$

*Author contributions.* J.R.M. and P.M.C. conceived the project. J.R.M. carried out the data analysis, wrote the paper and prepared the figures. K.Z., A.A. and C.H. gave much invaluable advice on analysis, mathematics and the general direction of the project as well as commented on the manuscript.

*Competing interests.* The authors declare that they have no conflict of interest

*Acknowledgements.* This work and its contributors (J.R.M., A.A., K.Z., C.H. and P.M.C.) were supported by the European Research Council (ERC) ECCLES project and by the Newton Fund through the Met Office Climate Science for Service Partnership Brazil (CSSP Brazil), also by a Faculty Research Grant awarded by the Committee on Research from the University of California, Santa Cruz (K.Z.) and the UK Centre of Ecology and Hydrology (CEH) National Capability Fund (C.H.).

We also wish to thank Ted Feldpausch for his many helpful comments and advice regarding Amazon forests, their allometry and analysis.

We particularly wish to thank the hard-working teams of researchers working to gather the RAINFOR data and share it through the Forest-Plots network. The principal investigators (PIs) who worked on each of the forest plots used that we wish to thank are Samuel Almeida, Esteban Álvarez Dávila, Luiz Aragão, Alejandro Araujo-Murakami, Luzmila Arroyo, Timothy Baker, Jorcely Barroso, Roel Brienen, Fernando





Cornejo Valverde, Maria Cristina Peñuela-Mora, William Farfan-Rios, Ted Feldpausch, Eurídice Honorio Coronado, Ben Hur Marimon Junior, Eliana Jimenez-Rojas Jon Lloyd, Yadvinder Malhi, Alexander Parada Gutierrez, Guido Pardo, Beatriz Marimon, Casimiro Mendoza, Irina Mendoza Polo, Abel Monteagudo-Mendoza, David Neill, Nadir Pallqui Camacho, Oliver Phillips, Nigel Pitman, Hirma Ramírez-Angulo, Freddy Ramirez Arevalo, Zorayda Restrepo Correa, Miles Silman, Javier Silva Espejo, Marcos Silveira, John Terborgh, Geertje van der Heijden, Rodolfo Vasquez Martinez, Emilio Vilanova Torre, Luis Valenzuela Gamarra and Vincent Vos.

Also see the supplementary material Table S1 for a more detailed list of which plot each PI worked on.



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
