# Peer review of "Validation of demographic equilibrium theory against tree-size distributions and biomass density in Amazonia"

_Biogeosciences, 2019_

## Referee Comment (RC1) · Anonymous Referee #1 · 27 Aug 2019

General comment I have mixed impressions of this manuscript. On the positive side, I thought that the exercise of fitting different models to the RAINFOR plot data was a worthwhile idea. It was informative to see the biases associated with the different models. However, the manuscript fell short in two general ways. First, the manuscript does not provide a deep explanation of what the results meant. In particular, the Discussion seemed superficial and primarily summarized the results. Second, by focusing on curve-fitting, the manuscript did not seem very creative. There were, perhaps, some missed opportunities for analysis. Several further suggestions follow.

Specific comments I think that this manuscript would be more impactful if it were or-

ganized around an explicit scientific question or hypothesis. In its current form, the manuscript is focused on the implicit question of whether DET or MST better fits the Amazon plot data. This implicit question strikes me as too technical. I would like to challenge the authors to develop a question that is more focused on the fundamental biology rather than a close-ended question of which model is better.

At the end of the paper, the authors discuss the work of Zhou and Lin (2018), who discuss a "fundament flaw" in the MST model. If the authors knew this, why did they bother with MST model at all in their own analysis? The way that the text is currently framed, one is left with the impression that the MST model was a straw man.

Artificial imposition of a maximum tree size seems unsatisfying to me. That it is needed suggests that there is a problem with the size-dependence of the mortality and/or growth rates. How might mortality (and/or growth) rates be modified so that maximum tree size would be a predictive outcome of the model?

Page 1, line 5: Here and elsewhere in the manuscript, it is stated that one model is "better" than another. But "better" in what sense? Blanket assertions that one model is better than another seem unwarranted to me.

Page 1, line 16: I did not see any whole-continent analysis.

Page 5, line 25: Please explicitly describe your algorithm.

Page 8, line 13: What is a "data point"? A stem? A size class? Something else?

Equation 14: I do not see how the second equality follows from the first. Please be more explicit.

Equation 15: What is S?

Page 9, lines 9-11: from the text, it looks like the two parameters were not estimated jointly: the parameter mu1 was estimated first, and then the estimate of mu1 was used to estimate phi. The problem with this procedure is that the estimation of mu1 itself

depends on phi, which is initially unknown. I am left confused about exactly what the authors did.

Page 9, line 19: The inequality is not sufficient to justify the assumption. Rather, the entire argument of the exponential must be small. For example, what if Dmax » DL, but c » mu1? I know it did not turn out that way, but it could have.

Page 10, line 24 through Page 11, line 1: It would help to justify this statement.

Page 27, lines 3-4: The text is misleading because the biomass was not actually observed.

Most of the Discussion is a re-statement of the Results. This is a major weakness of the manuscript because the significance of the results is left unexplained.

I found numerous typos (to list a few: page 3 line 9; Fig 1 x-axis label; page 10 line 22). The manuscript would benefit from a careful proofreading.

---

## Referee Comment (RC2) · Anonymous Referee #2 · 5 Nov 2019

This manuscript analyzes previously published data from forest censuses at 120 plots in the Amazon and evaluates their tree size distributions are fit by alternative models. Tree diameter distributions and tree biomass distributions are fitted with (1) left-truncated Weibull distributions with two free parameters, consistent with demographic equilibrium theory (DET) under the assumption of size-independent mortality and power-function growth; (2) left-truncated Weibull distributions with one free parameter, consistent with DET, size-independent-mortality, and growth that is a pre-specified power consistent with metabolic scaling theory for tree growth (1/3 power for diameter, $\frac{3}{4}$ for mass); and (3) metabolic scaling theory for size distributions (MST) meaning -2 power scaling for diameter distributions and -11/8 power scaling for biomass distributions. The distributions are fit for all plots combined, for geographic subsets of plots, and for individual plots. Biomass distributions are obtained by combining measured diameters with region-specific allometric equations for height, taxonomically assigned wood densities, and pantropical biomass equations based on diameter, height, and wood density. The models are fitted with maximum likelihood, and the alternative models are compared using AIC and BIC for the size distributions, as well as in terms of their ability to predict total AGB over all trees combined. The analysis is motivated in terms of the need to develop relatively simple models of size structure for global vegetation models.

The results show that size distributions are better fit by the DET-based models than by MST, and that the 2-fitted-parameter model is preferred to the 1-parameter model for the dataset as a whole whereas the 1-parameter model is preferred for most individual plots and regions. In the 2-parameter fits, the fitted parameter that is derived from the growth exponent is distributed around the values expected under MST growth theory. The two parameters of the 2-parameter fits are shown to be correlated across plots. The tree biomass distributions for these datasets are hump-shaped, reflecting the lower truncation of diameter at 10 cm combined with the variation in wood density among trees. Total plot biomass is reasonably well-predicted by the DET-based models, but not by the MST-based models, as the latter greatly overestimate biomass. The manuscript also contains an appendix that derives equations for the plot-level tree size distributions, total mass, and total mass growth from a certain set of assumptions, and that is referenced in the discussion.

The finding that MST is a poor fit and that Weibull functions are better fits for these tropical forest datasets is consistent with previous findings for other tropical forests, e.g., Muller-Landau et al. 2006, as cited. In some ways the finding that MST is a poor fit to size distributions seems like beating a dead horse at this point – the initial motivation for the MST size distribution argument always was a bit of a sleight of hand from even-aged stand self-thinning arguments used as a justification for uneven-aged stand

size distributions, as noted by multiple previous authors, and essentially every good analysis has found that MST is not a good fit to size distributions. At the same time, I'm a firm believer that we need more replication in ecology, and that solid analyses of new datasets should always be publishable, even if the findings are not qualitatively novel. The main novel elements are (1) different datasets, (2) exactly which models are compared, (3) analysis of biomass distributions in addition to diameter distributions, and (4) the derivation of whole plot biomass and productivity functions under DET in appendix. I think the application to new datasets in itself makes the analysis publishable, and the specific models compared here are a reasonable and interesting choice. I'm not convinced that it makes sense to analyze biomass distributions. I am intrigued by the derivation, but found the presentation lacking in material needed to understand it.

One of the novel elements is comparing the relative fit of truncated Weibull distributions with 2 free parameters vs. 1 free parameter. I have some suggestions regarding the implementation and interpretation of these results. Regarding implementation, the relative fit of the 1- vs. 2-parameter models (and even MST) appears to relate strongly to sample size. The largest datasets tend to provide support for the 2-parameter models, whereas the smaller datasets support the 1-parameter models. Similarly, the larger datsets appear to have more similar values of some of the parameters, with greater spread in the small parameter datasets. The results mention these patterns in the context of explaining some outliers and suggesting that some regional differences might be due to sample sizes in different plots. I suggest evaluating the role of sample size explicitly, by plotting the following vs. sample size (with sample size on log scale axis perhaps?): the AIC difference between the models, the BIC difference between the models, and the values of each of the parameters. I also recommend considering analyses of how these quantities vary with sample size in random subsamples of the full dataset (I wonder if the distribution of points in Figure 4 simply reflects the increasing spread of smaller sample sizes while following a constraint curve set by the overall distribution). Depending on what these figures reveal, it may or may not be worth including them in the main text and/or SI. Regarding interpretation, the relatively good

fit of the 1-parameter model is interpreted as support for the MST prediction regarding growth scaling with size. This seems to me to be a bit of a stretch, considering that data on growth are not analyzed here, and that any particular size distribution is consistent with an infinite combination of growth and mortality functions. The relevant size distribution parameter is equal to the growth exponent only if growth is a power function of size and mortality is size-independent, and reality deviates considerably from these assumptions (e.g., Muller-Landau et al. 2006, Coomes & Allen 2007).

Fitting the biomass distributions is clearly novel, but I'm not convinced it is very useful considering how the empirical biomass distributions are derived. As usual, individual tree biomasses are estimates based on allometric equations combining measured diameters, regional height-diameter allometries, taxonomically assigned wood densities, and an allometric equation for biomass based on diameter, height, and wood density. And then, the fits are the same sort of tests (MST vs DET) but with allometrically transformed derivations. Basically there is the same kind of data in both datasets, but the diameters are actually measured, while the biomasses are allometric estimates (see Clark and Kellner 2012). And the artefactual peak in the biomass distributions for these diameter-truncated datasets is problematic in terms of the fits (also in terms of using the resulting distributions to predict whole-forest biomass). The biomass distributions are used here to estimate whole-forest biomass, but the whole-forest biomass could instead be calculated from the diameter distributions by combining those pdfs with height-diameter allometries and mean wood densities. So in sum, the biomass distribution analyses seem to me to be largely redundant and inherently inferior, with all the objectives better met with analyses of the diameter distributions.

The derivation of closed form DET solutions is potentially neat, but it seems strange to put this in the discussion, and I found the explanation insufficient. It's stated that the derivation is made under the assumption of the perfect plasticity approximation, but a key variable in implementing the perfect plasticity approximation is the size at which individuals reach the canopy (and below which they are in the understory) and there is

nothing here about deriving this critical size. In fact, it seems that there is nothing in the understory and a large fraction of space is simply empty of vegetation, which doesn't make sense for a closed-canopy forest. Farrior et al. (2016) derive size distributions for canopy individuals, understory individuals, and the whole forest under the perfect plasticity assumption combined with a power function scaling for crown area. What is the relationship of the derivation here to that work (which is not cited here)?

Other specific comments

What is the motivation for calculating and reporting $n\_l$ in the tables? It is not a free parameter. Why should we care about it?

How exactly is whole plot biomass predicted – with what lower bound? (results in Table 8 and figure 9) Is this done with a lower bound equal to the peak of the biomass distribution, and if so, how is that peak defined exactly? Does the lower bound for prediction vary across plots, or is it fixed?

This manuscript refers to the usefulness of this approach for the "Robust Ecosystem Demography" model, but that model is not explained here, and is referenced only in a manuscript in preparation. If this model is going to be mentioned, it needs to be explained in more detail here (even if it were published, and especially given that it is not).

Page 1, line 30. Need to explain Demographic Equilibrium Theory more at first mention.

Equation 6. Having a comma as part of the subscript seems needlessly confusing. I recommend removing the comma.

P4 L19. Shouldn't the correction be for the largest tree mass possible, not the largest tree mass observed? The observed maximum is highly sensitive to sample size.

P5, L5-6. Actually, it's more a derivation of self-thinning, that is then declared to apply also to unevenaged stands.

L22. What is "mixed forest"?

L25. Why would plots with more data for smaller trees be excluded? As long as all trees above 10 cm are sampled, the data should be fine. Any plot sampled down to 1 cm will have a large proportion of measurements below 10 cm, but that doesn't mean the data for trees above 10 cm is problematic.

Table 1. I recommend moving this to SI, as it is simply a table of parameters repeated from another paper.

L10, last line. Why?

Figure 6. If the functions are fitted only to data above the threshold, then the fitted lines should not be extended below this threshold.

Page 25, line 11. That's not what I see in the supplemental figures. Figure S25 and S26 have the two S. Western curves apparently right on top of each other. (In general, please give specific figure numbers etc. when referencing supplemental materials.)

Figure 9. Why not include the MST predictions too, for comparison? Consider putting all the panels on log-log scales.

Page 29, line 20. The DET model does not inherently assume these things, that is just how it was implemented here.

Figure 10. What are the units of the x axis? Please give dbh range corresponding to a 1 kg tree, for reference.

Appendix A. Please give a complete set of assumptions here. In addition to what is stated, is mortality constant for all trees (regardless of canopy status) or is mortality 100% in the understory? Are growth rates the same power function of size for all trees, or only for canopy trees, with zero growth in the understory? I recommend adding parameters to the assumption list as well (e.g., give here the power function parameters for crown area scaling with tree mass). The only way I can understand the

canopy not being 100% full, would be if mortality in the understory is 100%, and the model operated in discrete time (so that gaps created by mortality were not immediately filled), but these assumptions are not stated.

References

Clark, D. B., and J. R. Kellner. 2012. Tropical forest biomass estimation and the fallacy of misplaced concreteness. Journal Of Vegetation Science 23:1191-1196.

Coomes, D. A., and R. B. Allen. 2007. Mortality and tree-size distributions in natural mixed-age forests. Journal Of Ecology 95:27-40.

Farrior, C. E., S. A. Bohlman, S. Hubbell, and S. W. Pacala. 2016. Dominance of the suppressed: Power-law size structure in tropical forests. Science 351:155-157.

Muller-Landau, H. C., R. S. Condit, K. E. Harms, C. O. Marks, S. C. Thomas, S. Bunyavejchewin, G. Chuyong, L. Co, S. Davies, R. Foster, S. Gunatilleke, N. Gunatilleke, T. Hart, S. P. Hubbell, A. Itoh, A. R. Kassim, D. Kenfack, J. V. LaFrankie, D. Lagunzad, H. S. Lee, E. Losos, J. R. Makana, T. Ohkubo, C. Samper, R. Sukumar, I. F. Sun, N. M. N. Supardi, S. Tan, D. Thomas, J. Thompson, R. Valencia, M. I. Vallejo, G. V. Munoz, T. Yamakura, J. K. Zimmerman, H. S. Dattaraja, S. Esufali, P. Hall, F. L. He, C. Hernandez, S. Kiratiprayoon, H. S. Suresh, C. Wills, and P. Ashton. 2006. Comparing tropical forest tree size distributions with the predictions of metabolic ecology and equilibrium models. Ecology Letters 9:589-602.

---

## Author Comment (AC1) · 17 Dec 2019

1. I think that this manuscript would be more impactful if it were organized around an explicit scientific question or hypothesis. In its current form, the manuscript is focused on the implicit question of whether DET or MST better fits the Amazon plot data. This implicit question strikes me as too technical. I would like to challenge the authors to develop a question that is more focused on the fundamental biology rather than a close-ended question of which model is better.

2. Most of the Discussion is a re-statement of the Results. This is a major weakness of the manuscript because the significance of the results is left unexplained.

[Figure]

Response to 1 & 2: In the revised manuscript (see attached supplement with just the revised intro and discussion) we have reduced the emphasis on the comparative performance of DET and MST, instead focusing on the implications of the DET fits to the data. We have introduced new histograms to show the range of best-fit parameter values across all 124 ForestPlots sites. These show that: (a) best-fit values for the exponents relating tree-size to growth-rate have mean and median values close to those predicted by Metabolic Scaling Theory; (b) when the growth exponent is fixed at the MST value, the remaining fitting parameter (which represents the ratio of mortality to growth) clusters strongly around a common value across the ForestPlot sites.

We also discuss the relationship between the fitting parameters $\varphi$ and $\mu 1$ as a possible life-history trade-off within forest plots, resulting in dominance of either live-fast die-young or grow-slow live-long strategies based on local conditions.

These findings, and their possible consequences, are now more prominent in heavily reworked versions of the Abstract, Discussion and Conclusions sections. Also, we now more clearly separate the use of MST to define the allometric relationship between tree size and growth-rate (West et al., 1997) for which we find some observational support, from the MST size-distribution (MSTF, West et al., 2009) for which we do not.

3. At the end of the paper, the authors discuss the work of Zhou and Lin (2018), who discuss a "fundament flaw" in the MST model. If the authors knew this, why did they bother with MST model at all in their own analysis? The way that the text is currently framed, one is left with the impression that the MST model was a straw man

Response to 3: In our revised manuscript we have significantly reduced the emphasis on the comparison of the DET and MSTF models, in favour of focusing on the implications of the DET fits across the 124 ForestPlots sites. We also feel there is no issue with replicating the result of others, that MSTF is a poor model, especially with a dataset we believe has not been tested this way before.

4. Artificial imposition of a maximum tree size seems unsatisfying to me. That it is

needed suggests that there is a problem with the size-dependence of the mortality and/or growth rates. How might mortality (and/or growth) rates be modified so that maximum tree size would be a predictive outcome of the model?

Response to 4: The largest tree size in any dataset is largely driven by the statistical effect of large trees being rarer and therefore appearing less often in datasets with smaller sample size. However, the reviewer raises an interesting point regarding the largest possible tree size. Trees cannot grow infinitely large due to physical constraints (mechanical, hydraulic etc) but it an open question as to whether mortality prevents trees reaching these limits. While this is somewhat outside the scope of this study, we see this as an interesting avenue for future study.

5. Page 1, line 5: Here and elsewhere in the manuscript, it is stated that one model is "better" than another. But "better" in what sense? Blanket assertions that one model is better than another seem unwarranted to me.

Response to 5: We have reduced the use of the word 'better' and made changes to be more explicit regarding the basis of each comparative statement in the manuscript.

6. Page 1, line 16: I did not see any whole-continent analysis.

Response to 6: We have modified the text throughout to change the term continent to "all plots". While all plots is loosely a continental scale, to be more precise we have made this change. As the abstract has been heavily rewritten the specific line mentioned no longer exists.

7. Page 5, line 25: Please explicitly describe your algorithm.

Response: There is no algorithm, seems that the line was not clear enough. Have now clarified this by changing the line last two lines of that paragraph to read: "The 124 selected plots all had a consistent lower cut-off in measurements at 10 cm trunk diameter. Two available upper montane plots with very few measurements above 10 cm were not included in the 124 plots used, as they did not have enough measurements

to allow a reliable fit."

8. Page 8, line 13: What is a "data point"? A stem? A size class? Something else?

Response: Have now clarified this by changing the line to read:- "where Di is tree trunk diameter measurement of stem i in the dataset."

9. Equation 14: I do not see how the second equality follows from the first. Please be more explicit.

Response: We have clarified this point, explaining that the second equality does not follow from the first, but instead describes the range of validity of the equation. The equation is not valid for phi = 1. We have changed the comma separating the equation and the equality to instead read "for".

10. Equation 15: What is S?

Response: Our apologies this was a typo. This should have been D not S - now corrected.

11. Page 9, lines 9-11: from the text, it looks like the two parameters were not estimated jointly: the parameter mu1 was estimated first, and then the estimate of mu1 was used to estimate phi. The problem with this procedure is that the estimation of mu1 itself depends on phi, which is initially unknown. I am left confused about exactly what the authors did.

Response: Text has been modified to clarify. Now reads: - "Substituting Eq. (16) into Eq. (17) creates a function only of c and therefore $\varphi$. This allows minimisation of -L in terms of $\varphi$ by using Brent's bounded algorithm (Brent, 1973a). Once the optimum $\varphi$ has been found then $\mu1$ can be calculated from equation 16. As equation 16 is included in the minimisation of -L, then it means we are in fact solving for both parameters at once and are finding the maxima of L. This algorithm was tested both with real data and data generated by computer from known LTWD distributions, by plotting the L values against $\varphi$ and $\mu1$, to confirm the maxima was found correctly.

Once the parameters $\mu 1$ and $\varphi$ are estimated, then this allows nL, the tree density per size class at DL, to be obtained from these parameters and the known quantities of the total number of trees N and the plot area A. This can be derived by integrating the equation for n (Eq. 4), to give: -"

12. Page 9, line 19: The inequality is not sufficient to justify the assumption. Rather, the entire argument of the exponential must be small. For example, what if Dmax Âż DL, but c Âż mu1? I know it did not turn out that way, but it could have.

Response: For c » $\mu 1$ that would imply either $\varphi$ being a highly negative value or $\mu 1$ being very small (suggesting either very low mortality or very high growth). Both these scenarios are unlikely, but we have added a clarifying statement to the text to acknowledge this. Now reads: - "For this study it was found that as Dmax » DL for most cases (and that c is never much larger than $\mu 1$ ), n L could assumed to be: -"

13. Page 10, line 24 through Page 11, line 1: It would help to justify this statement.

Response: Text has been modified to clarify. Now reads: - "This is because these equations only evaluate the mass up to but not including the trees with mass equal to the largest value in the dataset. Therefore, to comply with the definition above it is necessary to add the mass of the largest trees back into the total biomass.

As the large trees are so rare this correction will be equivalent to adding just one tree of the largest mass mmax in the dataset divided by A, the total area of plots in the dataset."

14. Page 27, lines 3-4: The text is misleading because the biomass was not actually observed.

Response: Have changed all references from "observed biomass" to "allometric biomass" to emphasise that this is the sum of masses obtained from observed trunk diameter measurements converted by allometry to mass.

15. I found numerous typos, to list a few: page 3 line 9; Fig 1 x-axis label; page 10 line

22

Response: All corrected.

Please also note the supplement to this comment:
https://www.biogeosciences-discuss.net/bg-2019-262/bg-2019-262-AC1-supplement.zip

[Figure]

**Fig. 1.**

[Figure]

**Fig. 2.**

---

## Author Comment (AC2) · 17 Dec 2019

1. I suggest evaluating the role of sample size explicitly, by plotting the following vs. sample size (with sample size on log scale axis perhaps?): the AIC difference between the models, the BIC difference between the models, and the values of each of the parameters. I also recommend considering analyses of how these quantities vary with sample size in random subsamples of the full dataset (I wonder if the distribution of points in Figure 4 simply reflects the increasing spread of smaller sample sizes while following a constraint curve set by the overall distribution). Depending on what these figures reveal, it may or may not be worth including them in the main text and/or SI.

[Figure]

Response: We have tried the suggested approach and found no correlation with sample size for the AIC/BIC and also the fitting parameters $\mu 1$ and $\varphi$. We did though find a strong correlation of the AIC/BIC of the MST model with sample size. Reinforcing the point already made that the MST model only appears a good model when the sample size is small. We have added a plot (S30) to the supplementary material showing this and referenced to it from the main text.

2. Regarding interpretation, the relatively good fit of the 1-parameter model is interpreted as support for the MST prediction regarding growth scaling with size. This seems to me to be a bit of a stretch, considering that data on growth are not analyzed here, and that any particular size distribution is consistent with an infinite combination of growth and mortality functions. The relevant size distribution parameter is equal to the growth exponent only if growth is a power function of size and mortality is size-independent, and reality deviates considerably from these assumptions (e.g., Muller-Landau et al. 2006, Coomes & Allen 2007).

Response: Our study shows that the DET model with power-law growth and constant mortality assumptions fits well over the large scale. While we acknowledge that other more complex models of growth and mortality may also fit the distribution, our point is that this simple model with allometry based on theoretical principles does very well over these large scales of interest in climate models/DGVMs. If repeated measurements of growth and mortality were available on a very large that would be the definitive test, but until then it is sensible to use a simple model in applications such DGVMs where parameter sparsity is very desirable.

We have acknowledged your points in the discussion: -

"The clustering of $\varphi$ results close to the value predicted by MST allometry (Niklas and Spatz, 2004; West et al., 2009) suggest two possibilities. Either that the clustering represents an underlying "basin of attraction" that is modified by local conditions (Price et al., 2007) or that plots do not meet the model assumptions of growth, mortality

and equilibrium somehow lead to this clustering. We cannot say for certain why the plots cluster close to the MST values but it does lead to intriguing future avenues of study. It was suggested (Coomes and Allen, 2009; Coomes et al., 2011) that light competition should modify the MST scaling of growth with size. This would mean that for trunk diameter the growth scaling power would vary with size and be greater than the predicted MST value of 1/3. For our regional fits the fitted power it was slightly larger than the MST value of 1/3 in most cases but for the individual forest plots the value was very close to MST with no clear bias. So our results cannot be taken as conclusive evidence of light competition modifying the growth scaling but neither are they completely inconsistent with it."

3. Fitting the biomass distributions is clearly novel, but I'm not convinced it is very useful considering how the empirical biomass distributions are derived. As usual, individual tree biomasses are estimates based on allometric equations combining measured diameters, regional height-diameter allometries, taxonomically assigned wood densities, and an allometric equation for biomass based on diameter, height, and wood density. And then, the fits are the same sort of tests (MST vs DET) but with allometrically transformed derivations. Basically there is the same kind of data in both datasets, but the diameters are actually measured, while the biomasses are allometric estimates (see Clark and Kellner 2012). And the artefactual peak in the biomass distributions for these diameter-truncated datasets is problematic in terms of the fits (also in terms of using the resulting distributions to predict whole-forest biomass). The biomass distributions are used here to estimate whole-forest biomass, but the whole-forest biomass could instead be calculated from the diameter distributions by combining those pdfs with height-diameter allometries and mean wood densities. So in sum, the biomass distribution analyses seem to me to be largely redundant and inherently inferior, with all the objectives better met with analyses of the diameter distributions.

Response: Knowing how well a model replicates biomass of a distribution is very important when using a model for climate applications. The main reason for modelling

forests dynamically in Earth System Models is to capture changes in land carbon storage that could affect the rate of climate change. So, an important question is whether the equilibrium model can model the biomass correctly. The effect of artefactual peak is removed by comparing the allometric derived biomass and theory predictions from the same lower bound corresponding to the peak. We are not suggesting the biomass model has an application in predicting biomass from field data but is a useful check on the accuracy of the model and more intuitive climate relevant measure than statistical measures of goodness of fit.

4. What is the motivation for calculating and reporting n_l in the tables? It is not a free parameter. Why should we care about it?

Response: We have removed these from the main text but kept those in the supplementary material with an edit to the captions in the supplementary material to explain nl is calculated not fitted.

5. How exactly is whole plot biomass predicted – with what lower bound? (results in Table 8 and figure 9) Is this done with a lower bound equal to the peak of the biomass distribution, and if so, how is that peak defined exactly? Does the lower bound for prediction vary across plots, or is it fixed?

Response: We have added a new paragraph in the Biomass results section (section 4.6) to explain this.

"The value of mP was used for the lower bound for calculating the predicted biomass in equations Eq. (6), Eq. (7) and Eq. (10). The same values of mP were used to truncate the data when finding the biomass density. So, comparisons between the theory and the mass obtained directly from a combination of observation and allometry were always using the same lower truncation point for each dataset but varied between datasets. The values of mP used are given in Table 4 and the methodology used to estimate mP is in section 4.1.1."

And we have added a new sub section to the Mass Distribution results section (section 4.1): -

"When working with mass data the peak was eliminated from fitting by creating 40 bin edges (39 bins) in log-space (base e) from the smallest to largest tree in the dataset. These edges define the range of each bin and the value of each bin was selected as the midpoint in log-space. The data was then binned following these bins. Once the data was binned, the bin with the highest frequency was identified. The value of this bin was then used as the truncation point for the dataset when fitting to the dataset distribution. The binning was purely used to identify the peak and for plotting the data and not used during the MLE fitting process."

6. Page 1, line 30. Need to explain Demographic Equilibrium Theory more at first mention.

Response: We assume the reviewer is referring to Page 2, Line30. The intro has changed but the first mention of DET now reads:

"We follow Demographic Equilibrium Theory (DET) (Muller-Landau et al., 2006b) in assuming that forests are in a steady-state with size distributions completely determined by size-dependent functions of tree growth and mortality."

7. Equation 6. Having a comma as part of the subscript seems needlessly confusing. I recommend removing the comma.

Response: The notation to represent mass mortality to growth ratio is now $\mu\_m1$ without the comma and has been changed throughout the paper.

8. P4 L19. Shouldn't the correction be for the largest tree mass possible, not the largest tree mass observed? The observed maximum is highly sensitive to sample size.

Response: The objective is to correct for the maximum tree size based on the sample, so sensitivity to sample size is intended. In particular, we wanted to make sure the reverse CDF plots (in the supplementary material) matched for the largest tree size,

meaning the difference in biomass density at the lower bound (DL and mP) would represent the goodness of fit regardless of sample size effects. Maximum possible tree size is not something we know of, certainly we could not find anything published on this. So we still feel this is the best solution available to us.

9. P5, L5-6. Actually, it's more a derivation of self-thinning, that is then declared to apply also to uneven aged stands.

Response: Added a mention of self-thinning to this line. P5 First paragraph now reads: -

"Metabolic scaling theory is a theory of scaling of organisms with size, based on theories of metabolism, physics and chemistry (West, 1997; Muller-Landau et al., 2006a). This theory uses the predictions of the scaling of individuals to predict the larger scale patterns and structure of populations and communities. For forests this is in the form of using the scaling of photosynthesis of trees and the vascular structures that transport water to predict individual scaling. This is then combined with assumptions from self-thinning about how trees fill space to describe the expected forest size-distribution (Coomes, 2003; West et al., 2009). This leads to a power law distribution for trunk diameter: -"

10. L22. What is "mixed forest"?

Response: This has been clarified by adding in brackets after the term mixed forest: - "(not monoculture)"

11. L25. Why would plots with more data for smaller trees be excluded? As long as all trees above 10 cm are sampled, the data should be fine. Any plot sampled down to 1 cm will have a large proportion of measurements below 10 cm, but that doesn't mean the data for trees above 10 cm is problematic.

Response: The point was these plots had very few measurements above 10 cm so by applying the consistent 10 cm cutoff point have too few measurements to be useful.

Line now reads: - "Two available upper montane plots with very few measurements above 10 cm were not included in the 124 plots used, as they did not have enough measurements to allow a reliable fit."

12. Table 1. I recommend moving this to SI, as it is simply a table of parameters repeated from another paper.

Response: Done. Table 1 is now Table S1 in the supplementary material.

13. L10, last line. Why?

Response: We assume this is referring to Page 10 last line? This paragraph modified to say: -

"A correction term is added to Eq. (7) and Eq. (10) to make sure the biomass density correctly evaluates at the upper boundary (the mass of the largest tree mmax). This is because these equations only evaluate the mass up to but not including the trees with mass equal to the largest value in the dataset. Therefore, to comply with the definition above it is necessary to add the mass of the largest trees back into the total biomass.

As the large trees are so rare this correction will be equivalent to adding just one tree of the largest mass mmax in the dataset divided by A, the total area of plots in the dataset. "

14. Figure 6. If the functions are fitted only to data above the threshold, then the fitted lines should not be extended below this threshold .

Response: Changed as requested (see attached figure 6).

15. Page 25, line 11. That's not what I see in the supplemental figures. Figure S25 and S26 have the two S. Western curves apparently right on top of each other. (In general, please give specific figure numbers etc. when referencing supplemental materials.)

Response: We have added the figure numbers to the text and we have also added an extra line to this paragraph regarding the S.Western region: "Interestingly, two regions

(S.Western and Ecuador) had a worse fit for two parameter DET-LTWD. The S.Western region though fits the biomass within 2% regardless of the choice of upper bound or DET model, so the very slight difference in the biomass density prediction is almost certainly not significant for this region. When the reverse cumulative biomass density, defined as biomass density of all trees above a given tree mass, is plotted for Ecuador (see supplementary material Figures S27 and S28) the error comes from the shape of the tail of the distribution, which is much flatter than theory. This could be due to it being a region with a smaller number of trees (4159) or could be due to higher mortality for large trees in this region."

16. Figure 9. Why not include the MST predictions too, for comparison? Consider putting all the panels on log-log scales.

Response: We have added a new MST subpanel to figure 9 as suggest (see attached figure 9).

17. Page 29, line 20. The DET model does not inherently assume these things, that is just how it was implemented here.

Response: The discussion was totally rewritten so this line no longer exists. The new first line of the discussion reads: - "In this paper we show that the Left-Truncated Weibull (LWTD), which is consistent with the Demographic Equilibrium Theory (DET) when the mortality is size independent and the growth is a power-law of tree size"

18. This manuscript refers to the usefulness of this approach for the "Robust Ecosystem Demography" model, but that model is not explained here, and is referenced only in a manuscript in preparation. If this model is going to be mentioned, it needs to be explained in more detail here (even if it were published, and especially given that it is not).

Response: Although the RED model is one motivation for us to carry-out this study, it is far from the only one. We also wish to understand the current size-distribution of

forests, regardless of the potential use of that understanding in that development of a new DGVM. In the revised manuscript we have therefore reduced the emphasis on RED as a motivating factor.

19. Figure 10. What are the units of the x axis? Please give dbh range corresponding to a 1 kg tree, for reference.

20. The derivation of closed form DET solutions is potentially neat, but it seems strange to put this in the discussion, and I found the explanation insufficient. It's stated that the derivation is made under the assumption of the perfect plasticity approximation, but a key variable in implementing the perfect plasticity approximation is the size at which individuals reach the canopy (and below which they are in the understory) and there is nothing here about deriving this critical size. In fact, it seems that there is nothing in the understory and a large fraction of space is simply empty of vegetation, which doesn't make sense for a closed-canopy forest. Farrior et al. (2016) derive size distributions for canopy individuals, understory individuals, and the whole forest under the perfect plasticity assumption combined with a power function scaling for crown area. What is the relationship of the derivation here to that work (which is not cited here)?

21. Appendix A. Please give a complete set of assumptions here. In addition to what is stated, is mortality constant for all trees (regardless of canopy status) or is mortality 100% in the understory? Are growth rates the same power function of size for all trees, or only for canopy trees, with zero growth in the understory? I recommend adding parameters to the assumption list as well (e.g., give here the power function parameters for crown area scaling with tree mass). The only way I can understand the canopy not being 100% full, would be if mortality in the understory is 100%, and the model operated in discrete time (so that gaps created by mortality were not immediately filled), but these assumptions are not stated.

Response: These three points (19-21) all refer to the RED model, a small part of which was included in this paper. As the main RED paper (Argles 2019) has now reached

public discussion in Geoscientific Model Development we have decided to remove all sections relating to RED from this paper. We now ask the reviewers and editors to refer to this paper. We have modified the introduction and discussion to reflect these changes. This includes the closed form solutions that in retrospect, are an add-on that are not required for the rest of this study.

References

Argles, A. P. K., Moore, J. R., Huntingford, C., Wiltshire, A. J., Jones, C. D., and Cox, P. M. 2019. Robust Ecosystem Demography (RED): a parsimonious approach to modelling vegetation dynamics in Earth System Models, Geoscientific Model Development, https://doi.org/10.5194/gmd-2019-300,.

———————————————————

[Figure]

**Fig. 1.** Supplement Figure S30

[Figure]

**Fig. 2.** Revised Figure 6

**Fig. 3.** Revised Figure 9

---

## Author Response (AR1)

**Reply to Editors Report**

> **Editors Report:** Thank you for providing author comments in response to the referee comments. Both referees evaluated that your manuscript has good scientific quality and fair to good scientific significance and presentation quality. They acknowledged that your study topic is timely and fits to the journal scope, and recommended to make further data-model analyses with clearer explanations. The tree-size distribution model with a fewer number of parameters would help constrain DGVMs.
> I studied the referee comments, your responses, and revised Introduction and Discussion and conclude that the manuscript would be adequately improved in clarity and insightfulness. I am looking forward to receiving the revised manuscript.

**Authors' Response:**

Thank you for your report and agreement with the changes we proposed in our response to the referee's comments (previously submitted, please refer to them for list of major changes). Please find in this submission the updated manuscript, supplementary material and in this document the tracked changes version of the manuscript compared to that submitted in July.

The changes made are extensive, as we had proposed, and note that Most figures have changes that cannot be shown via latexdiff tracked changes and also note that the figure 10 in the previous version has been cut, again this cannot be shown in the tracked changes, so we are highlighting these changes here. Also note the previous Table 1 (the height and mass allometry coefficients used to convert trunk diameter to mass) has been moved to the supplementary material.

We have also gone through the manuscript and tried to eliminate any remaining typographic errors. We now hope these have all been found and removed. If you or the journal editorial team have any further comments or queries do not hesitate to contact us.

**Changes to Figures: -**

Figure 1 – Corrected longitude spelling on x axis label.

Figure 4 swapped with figure 5

New Figure 4 – Added second panel b) showing the histogram of fitted $\mu_1$ values for both 1 and 2-parameter fits. The original panel a) was rebinned to match the number of bins in the new panel b).

New Figure 5 – Compared to its previous form as the old figure 4 the only change is the addition of the line showing equation 26 fits the trend of the points relating the two fitting parameters.

Figure 6 – Restricted the fitting lines to the range of tree masses used in the fitting process. This is to more clearly show that trees with mass below $m_P$ were excluded from the fit.

Figure 7 – Again added a line to show the fit of equation 26 to the relationship between fitting parameters.

Figure 8 - Added second panel b) showing the histogram of fitted $\mu_{m1}$ values for both 1 and 2-parameter fits. The original panel a) was rebinned to match the number of bins in the new panel b).

Figure 9 – Added extra panel for MST biomass and changed the x-axis label from "observed biomass" to "allometric biomass".

Figure 10 - Removed

[revised manuscript text omitted]

---

## Author Response (AR2)

**Reply to Editors Report**

>**Editors Comment:** Thank you for uploading the revised manuscript, in which extensive changes were made. I confirmed that your made appropriate amendments to the manuscript, especially in terms of model description and presentations. Therefore, I conclude that the manuscript is acceptable for publication (please look technical point).

>Technical point

>Please check the reference list again. For example, Argles et al. (2019) is published as a discussion paper as Longo et al. (2019). For several papers you show both URL and DOI (e.g. Feldpausch et al. 2011, 2012; Kohyama et al. 1991, 2003). Also, please consistently use capital for the first letters of journal names (e.g. Coomes et al. 2003).

**Authors' Response:**

Thank you for your comment and for accepting our paper for publication.

We have modified the manuscript in line with the Editor's technical note, these can be seen in the marked-up manuscript that is part of this document.

Changes to References (as requested):-

- Argles 2019 – Journal title changed from "GMD" to "GMD Discussions"
- Coomes 2003 – Journal title changed from "Ecology letters" to "Ecology Letters"
- Longo 2019 – Journal Title changed from "GMD Discussions" to "GMD" and updated the doi as this paper has now reached full publication.
- We also removed the URLs from Feldpausch 2011, 2012, Moore 2018 and Shugart 2018, as indicated by the editor these also have DOIs and do not need both.  Note we did not see URLs for the Kohyama papers, as mentioned in the technical point above, so could not remove them, if this point was in error and you wanted different changes please get back to us.
- We also corrected the DOIs (as they had "https:\\doi.org\https:\\doi.org" rather than the correct "https:\\doi.org" ie repeated twice) for Brienen 2015, Coomes 2003, Coomes 2009, Cox 2000, Feldpausch 2011, Fisher 2018, Friedlingstein 2014, Kohyama 1991, Kohyama 2003, Lima 2016, Moorcroft 2001, Muller-Landau 2006b, Niklas and Spatz 2004, Taubert 2013, West 2009, White 2008, Zanne 2009.

Further changes made (minor typographic):-

- Page 8, Line 18 – Minor clarification.
- Page 9, Line 4 – Minor clarification.
- Page 9, Line 9 – Added missing reference to supplementary figures.
- Page 19 – Various minor changes for clarity or to remove repetition.
- Page 20 line 5 - Removed extra full stop that should not have been there.
- Page 31 line 21 - Added the word "small", as this was missing and should have been there.
- Page 33 line 12 - Clarified whether text was referring to 1 or 2 parameter DET model.
- Added missing items to the legends of Figures 5 and 7, no changes to plot itself, just the legend to meet the manuscript preparation guidelines that require all symbols to be in the legend.

[revised manuscript text omitted]